# HAI-Eval: Measuring Human-AI Synergy in Collaborative Coding

## Abstract

LLM-powered coding agents are reshaping the development paradigm. However, existing evaluation systems, neither traditional tests for humans nor benchmarks for LLMs, fail to capture this shift. They remain focused on well-defined algorithmic problems, which excludes problems where success depends on human-AI collaboration. Such collaborative problems not only require human reasoning to interpret complex contexts and guide solution strategies, but also demand AI efficiency for implementation. To bridge this gap, we introduce `HAI-Eval`, a unified benchmark designed to measure the synergy of human-AI partnership in coding. HAI-Eval's core innovation is its "Collaboration-Necessary" problem templates, which are intractable for both standalone LLMs and unaided humans, but solvable through effective collaboration. Specifically, HAI-Eval uses 45 templates to dynamically create tasks. It also provides a standardized IDE for human participants and a reproducible toolkit with 450 task instances for LLMs, ensuring an ecologically valid evaluation. We conduct a within-subject study with 45 participants and benchmark their performance against 5 state-of-the-art LLMs under 4 different levels of human intervention. Results show that standalone LLMs and unaided participants achieve poor pass rates (**0.67%** and **18.89%**), human–AI collaboration significantly improves performance to **31.11%**. Our analysis reveals an emerging co-reasoning partnership. This finding challenges the traditional human-tool hierarchy by showing that strategic breakthroughs can originate from either humans or AI. HAI-Eval establishes not only a challenging benchmark for next-generation coding agents but also a grounded, scalable framework for assessing core developer competencies in the AI era. Our benchmark and interactive demo are openly accessible.

## 1 Introduction

Coding agents powered by Large Language Models (LLMs) are fundamentally reshaping the software development paradigm (Soni et al., 2023; Coutinho et al., 2024; Martinović & Rozić, 2025). Tools such as Claude Code (Anthropic, 2024), Cursor (Anysphere, 2024), and GitHub Copilot (GitHub, 2024) are now widely used in practice (Perumal, 2025). As a result, the role of a developer is shifting from that of a code producer to a leader within a human-AI collaborative system. Developers are now responsible for strategic planning, directing AI contributions, and ensuring final code quality (Alenezi & Akour, 2025; Eshraghian et al., 2025). Simultaneously, coding agents are evolving to automate increasingly higher-level tasks. This trend continuously extends the frontier of human-AI collaboration (Hou et al., 2024; Nghiem et al., 2024; Pezzè et al., 2025).

Nonetheless, this revolution in development practice exposes a fundamental gap in evaluation. Most current assessments, for both humans and AI, share a common flaw: they assume the existence of a *perfectly defined problem*. Human-focused platforms like LeetCode (LeetCode, 2015) and Codeforces (Codeforces, 2010), emphasize well-structured algorithmic problems, which incentivize developers to master skills that are increasingly automated (opentools, 2025; April Bohnert, 2023). Similarly, recent AI benchmarks that aim for realism (Jimenez et al., 2024; Yu et al., 2024; Li et al., 2024b) often focus on environmental details (e.g., using real-world repositories), but they still frame tasks as cleanly defined problems. These benchmarks overlook the complex stage of problem formulation and thus fail to evaluate higher-order reasoning skills. Such skills, including *problem formulation*, *requirement engineering*, and *strategic decomposition*, are essential for navigating ambiguity before a problem is fully defined (Hemmat et al., 2025; Mozannar et al., 2024a).

Some advanced evaluation methods, such as LLM-as-a-Judge (Zheng et al., 2023; Li et al., 2024a) and Agent-as-a-Judge (Zhuge et al., 2024), are emerging to evaluate performance on higher-order definitions. However, this progress in evaluators has not been matched by an evolution in datasets; these powerful methods are still applied to benchmarks with only perfectly defined problems (Wang et al., 2025; Crupi et al., 2025), limiting the comprehensive evaluation of above crucial skills.

This situation highlights two critical needs: a framework for quantifying the human contribution in human-AI collaboration (Haupt & Brynjolfsson, 2025), and a benchmark that pushes LLMs toward higher-order reasoning skills as human experts have (Zhang et al., 2024). To address both needs, we introduce `HAI-Eval`, a unified benchmark designed to measure human-AI synergy in collaborative coding through "collaboration-necessary" tasks. Figure 1 provides an overview of the HAI-Eval workflow. HAI-Eval comprises four core components: (i) A problem template bank with 45 templates spanning 3 professional tracks and 3 difficulty levels. Each template is designed to be intractable for either state-of-the-art (SOTA) LLMs or unaided developers, thus allowing the measurement of performance gains driven by collaboration. (ii) An agentic task system dynamically creates unique, context-rich tasks from these problem templates. (iii) An ecologically valid cloud Integrated Development Environment (IDE), built upon Copilot and VS Code (Microsoft, 2015), provides a realistic, full-featured interface for human evaluation. (iv) An autonomous evaluation toolkit with a custom VS Code extension and 450 manually-curated task instances provides a reproducible interface for benchmarking LLMs.

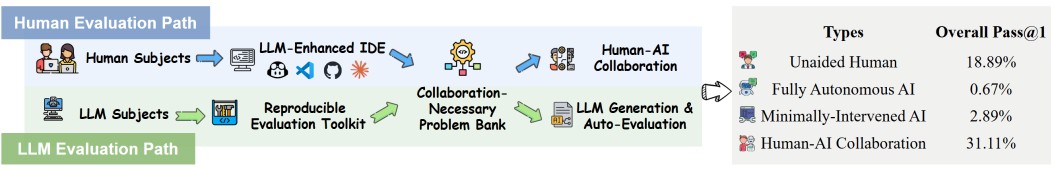

Figure 1: HAI-Eval provides two interfaces for evaluation, underscoring its two contributions to the community. The chart displays the performance improvement by human-AI collaboration.

Leveraging `HAI-Eval`'s dual interfaces, we conduct a comprehensive empirical user study. Our human evaluation involves a within-subject study design with 45 expert users, proficient in both unaided and AI-assisted coding. We benchmark their performance against 5 SOTA LLMs under varying levels of human intervention. To ensure reliability, we validate results through both participant feedback and independent expert review. The results demonstrate that HAI-Eval effectively quantifies the contribution of human-AI collaboration compared to standalone developers and LLMs (see Figure 1). More importantly, our findings show a shift in how experts use coding agents. The agent is no longer just an implementation tool but becomes a partner in strategic design. This new dynamic of human-AI co-reasoning provides a clear direction for future developer education and research into fully autonomous coding agents.

Our core contributions are summarized as follows:

❶ *Unified Benchmark.* We introduce `HAI-Eval`, the first benchmark designed both to quantify the human contribution in AI-assisted coding and to challenge the higher-order reasoning ability of coding agents with human-AI collaboration-dependent coding tasks. It consists of a carefully curated problem bank and an agentic dynamic task instantiation system.

❷ *Dual Interfaces.* For human evaluation, we build a cloud IDE integrated with coding agents, offering an authentic development experience. For benchmarking LLMs, we release a reproducible toolkit with 450 manually-curated static tasks.

❸ *Empirical Validation.* Our experiments quantify the contribution of human developers, expose the limitations of current SOTA coding agents, and derive key insights for future research.

## 2 RELATED WORKS

**Assessment of Human Developers.** Currently, the technical evaluation of developers is defined by recruitment-oriented platforms like LeetCode, HackerRank (HackerRank, 2012), and TopCoder (TopCoder, 2001), as well as competitive programming platforms like Codeforces and Luogu (Luogu, 2013). By placing developers in an isolated, well-defined setting with clear inputs and outputs, these platforms enable standardized and reproducible assessment. However, their core philosophy of evaluating a developer in isolation as a mere code producer is fundamentally misaligned with the modern AI-assisted paradigm, where value is generated through the synergistic

partnership between developers and coding agents. Consequently, they fail to measure the competencies that now define engineering excellence: interpreting ambiguous requirements, leveraging advanced development tools, strategically collaborating with coding agents, and critically validating AI-generated solutions (Chen et al., 2024; Priya R, 2025).

**Benchmarks for Coding Agents.** Numerous benchmarks have emerged to evaluate the coding performance of LLMs and agents. Foundational benchmarks like HumanEval (Chen et al., 2021) and MBPP (Austin et al., 2021) test function-level code generation, while more sophisticated ones, including SWE-Bench (Jimenez et al., 2024), MLE-Bench (He et al., 2024), ClassEval (Du et al., 2023), and DS-1000 (Lai et al., 2023), challenge agents to resolve real-world engineering problems in specific domains. Despite their increasing realism, these benchmarks evaluate agents on well-defined, pre-specified tasks with complete requirements. Even stress-test benchmarks designed to push the performance limits such as LiveCodeBench series (Zheng et al., 2024; 2025) are largely one-dimensional, focusing on extreme algorithmic complexity. Consequently, they systematically fail to evaluate higher-order skills such as requirement engineering, navigating ambiguity in under-specified tasks, and formulating executable plans from complex scenarios.

**User Studies in AI-Assisted Coding.** A growing body of user studies has examined the impact of AI-assisted coding in real-world settings. Fundamentally, this line of research investigates a crucial initial question: "Does AI make developers *faster*?" Numerous enterprise-based experiments (Cui et al., 2025; Paradis et al., 2025; Bakal et al., 2025) and academic investigations (Dohmke et al., 2023; Solohubov et al., 2023; Prather et al., 2024; Barke et al., 2023) report substantial productivity gains ranging from qualitative insights to quantitative improvements of 15%–30%, while some studies reveal negative impacts on efficiency (Vaithilingam et al., 2022; Becker et al., 2025). These conflicting findings arguably stem from a methodological limitation: relying on ad-hoc evaluations in specific settings and thus lacking the standardized, benchmark-style reproducibility for generalizable conclusions. This limitation also prevents the community from addressing a more profound question systematically: "Does AI make us more *capable*?" This question subsumes the limitations of prior work, focusing not on speed metrics for general tasks, but on the ability to solve previously intractable problems that explicitly isolate human contributions at the frontier of AI capabilities. Although recent initiatives have begun to explore frameworks for evaluating human-agent interaction (Lee et al., 2022; Shao et al., 2024), a standardized benchmark designed for the complex domain of coding remains a critical gap.

## 3 DESIGN PRINCIPLE OF HAI-EVAL

Drawing on distributed cognition theory (Hutchins, 1995; Hollan et al., 2000), which conceptualizes cognition as a process distributed across humans, artifacts, and environments, and on established frameworks for human-AI collaboration (Bansal et al., 2019; Amershi et al., 2019; Shneiderman, 2020), the design of HAI-Eval is anchored in two foundational principles: **Ecological Validity** and **Necessary Collaboration**. Together, these two principles instantiate distributed cognition: ecological validity ensures that cognition is evaluated in authentic, tool-mediated settings, while necessary collaboration ensures that the human–AI pair, not either alone, is the operative unit of analysis. This grounding not only justifies our design but also sets a research agenda for benchmarks that capture the true dynamics of human–AI problem solving.

**Ecological Validity.** *Ecological validity ensures that the skills being evaluated, whether human or AI, are meaningful and transferable to professional contexts.* This principle mandates that the test environment must faithfully mirror the real-world workflow, drawing on the ecological validity theory in psychology (Holleman et al., 2020; Kihlstrom, 2021; Ullah et al., 2023) and evidence from software engineering research (Fragiadakis et al., 2024; Sergeyuk & Zaytsev, 2025). Accordingly, HAI-Eval formalizes ecological validity along three dimensions:

❶ **Interaction Method (Human→Machine):** The user's interaction must be analogous to industry standards, e.g., a VS Code-like workflow. The focus is on preventing the introduction of confounding variables related to tool proficiency.

❷ **Assistance Strength (Machine→Human):** The support provided by the coding agent must reflect current professional practice. This requires it to be powered by SOTA LLMs.

❸ **Task Requirement (Task→Human-Machine):** Tasks must simulate a real-world project context. First, the task must represent an authentic engineering scenario. Second, requirements should be presented in a natural format.

**Necessary Collaboration.** *Necessary collaboration defines a problem space where neither humans nor AI can achieve optimal results independently, thereby enabling the quantification of collaborative value beyond simple productivity metrics.* This principle stipulates that tasks must be designed to necessitate genuine human-AI partnership. It builds on theoretical advances in complementary human-machine collaboration (Donahue & Kleinberg, 2022) and human-AI team performance (Vaccaro et al., 2024; Fujikawa et al., 2024). Distributed cognition reinforces this principle by framing the human–AI pair as the operative cognitive system where performance emerges only when both contribute complementary strengths. HAI-Eval operationalizes this through two constraints:

❶ **AI-Incomplete Tasks:** Tasks must include elements that remain intractable for SOTA coding agents, thus requiring human guidance. This intractability should be introduced through real-world contextual complexity rather than algorithmic difficulty, ensuring that tasks expose a fundamental gap between current LLMs and developers, namely, the ability to perform higher-order reasoning (Rangarajan et al., 2024; Xin et al., 2024), which we ground in Relational Complexity Theory (Halford et al., 1998).

❷ **Human Reliance on AI:** Tasks must make purely manual solutions suboptimal, encouraging strategic collaboration where humans recognize and execute optimal task delegation to AI partners. This reliance should be created by incorporating scale, repetition, or low-level implementation demands that make manual completion inefficient, ensuring that tasks test the ability to orchestrate collaboration and integrate AI-generated outputs (Hemmer et al., 2025).

Formally, they impose the following constraints on any given task $T$:

$$\Pr(\mathrm{Solve}(t, \mathcal{A})) \leq \theta_{\mathrm{low}}; \quad \mathbb{E}[\mathrm{Score}(s_{\mathcal{H}+\mathcal{A}})] - \mathbb{E}[\mathrm{Score}(s_{\mathcal{H}})] \geq \delta \tag{1}$$

where $\theta_{\mathrm{low}}$ represents the success probability for any autonomous agent $\mathcal{A}$, which should be negligible, $\delta > 0$ defines a significant collaboration incentive margin, and $\mathbb{E}[\mathrm{Score}(\cdot)]$ denotes the expected overall performance score for a solution submitted by the human $\mathcal{H}$, the agent $\mathcal{A}$, or the team $\mathcal{H}+\mathcal{A}$.

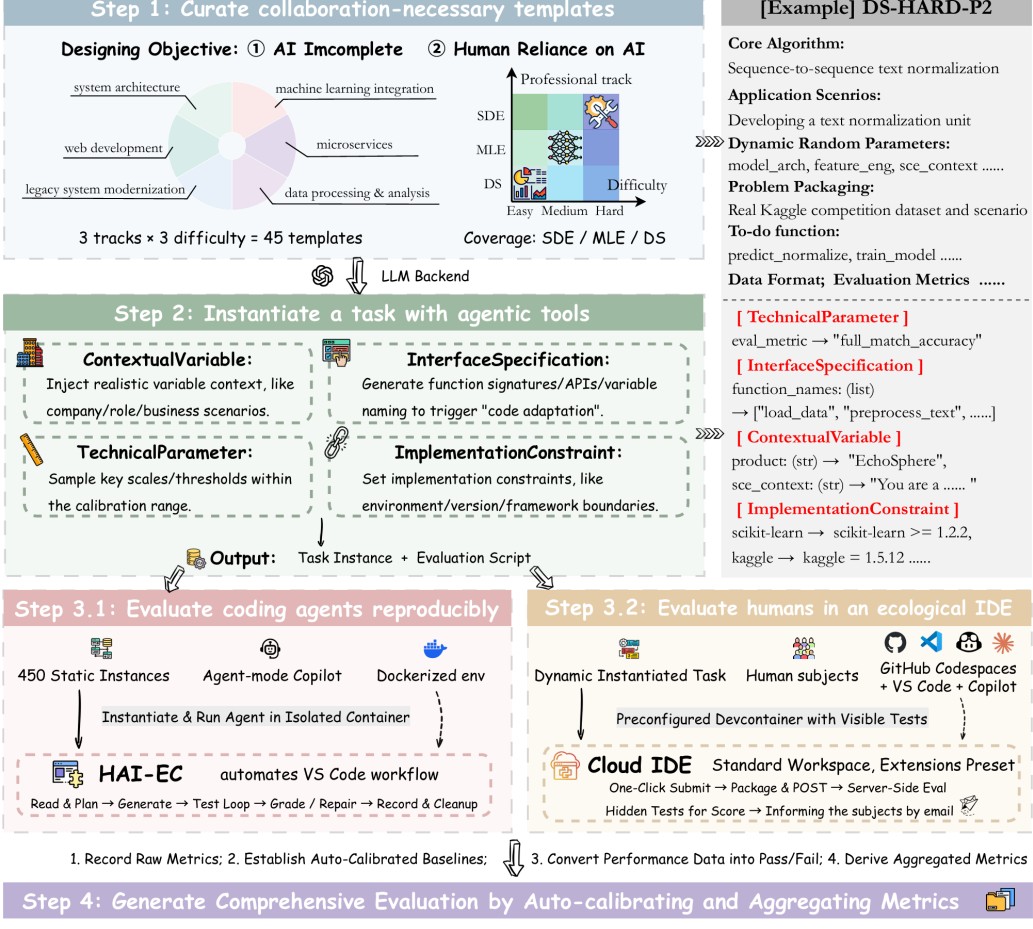

Figure 2: The overall architecture of HAI-Eval.

## 4 HAI-EVAL FRAMEWORK

Figure 2 illustrates the architecture of **HAI-Eval**, the implementation of our design principles. Its four core components work synergistically to enable comprehensive evaluation of both developers and coding agents. The **Problem Template Bank** and **Agentic Task System** are designed to uphold both Ecological Validity and Necessary Collaboration, while the **Standardized Cloud IDE** and **Evaluation Toolkit** primarily ensure Ecological Validity by providing authentic and reproducible evaluation environments. This section also details the evaluation metrics used by HAI-Eval.

### 4.1 PROBLEM TEMPLATE BANK

The central design challenge for our problem bank is to construct tasks that are simultaneously intractable for SOTA LLMs and suboptimal for unaided human developers, yet remain solvable through effective human-AI collaboration. Our problem template bank is engineered to meet this challenge by wrapping basic algorithmic cores with layers of complexities that require human reasoning to resolve (Figure 3). To ensure broad coverage across contemporary development scenarios, the bank contains 45 templates arranged in a 3×3 matrix spanning three difficulty levels and three professional tracks: Software Development Engineer (SDE), Machine Learning Engineer (MLE), and Data Scientist (DS). Details on how difficulty levels are determined and calibrated are provided in Appendix D. This design is further operationalized through two complementary approaches, each targeting the limitations of one side in the collaboration:

➡ *AI-Incomplete.* To ensure templates are AI-Incomplete, we introduce relational complexities (Halford et al., 1998) that impede LLM comprehension by strategically injecting elements designed to prevent LLMs from inferring well-defined specifications and directly decomposing them into actionable steps, thus making human intervention, or advanced LLM reasoning capabilities, essential for successful task completion. These relational complexities are drawn from documented challenges in real-world software development. This includes (i) underspecified requirements that require human-level clarification and decomposition (Kamsties et al., 2001; Mozannar et al., 2024b), (ii) multimodal specifications, such as UML or ER diagrams, which require extracting logic and constraints from information-dense symbolic systems (Larkin & Simon, 1987; Siau & Cao, 2001; Ozkaya & Erata, 2020), (iii) legacy codebases with minimal documentation (Jimenez et al., 2024), and (iv) domain-specific constraints embedded in business logic (Joel et al., 2024; Gu et al., 2025).

➡ *Human Reliance on AI.* To foster human reliance on AI, we employ a complementary strategy of injecting elements that make purely manual solutions prohibitively inefficient and thus encourage humans to recognize and execute optimal task delegation to AI partners. This strategy is based on established practices in human-AI collaboration (Hemmer et al., 2023), and achieved through two methods: (i) embedding components that are tedious or repetitive to implement manually, such as boilerplate code, data parsing scripts, and configuration-heavy setup tasks, coupled with time-sensitive scoring to render manual solutions suboptimal (Pandey et al., 2024; Kuutila et al., 2020); and (ii) incorporating industry-relevant algorithms or APIs that are practically valuable but not typically covered in standard curricula, creating strategic knowledge gaps that encourage human developers to leverage LLMs as specialized tools for subproblems (Nam et al., 2024).

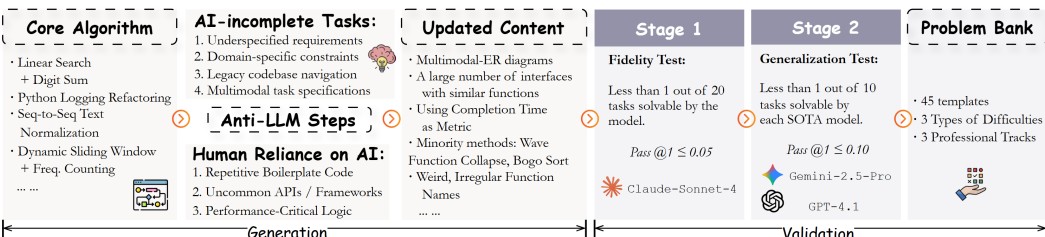

Figure 3: The design-validation pipeline for transforming algorithmic cores into templates.

To rigorously validate that each template is *AI-Incomplete*, we implement a two-stage validation protocol, detailed in Appendix B. The first stage is a **fidelity test** against a widely deployed commercial baseline model ($\mathcal{A}_{\text{base}}$). The second stage is a **generalization test** across a representative set of state-of-the-art coding LLMs ($\mathbb{A}_{\text{SOTA}}$). Formally, let $T$ denote a problem template, and let $S(T)$ and $S'(T)$ be sets of 20 and 10 task instances from $T$, respectively. For an agent $\mathcal{A}$ and task $t$, let $\text{Pass}(\mathcal{A}, t) \in 0, 1$ indicate whether $\mathcal{A}$ solves $t$ in one attempt. A template $T$ is accepted only if it

satisfies both of the following benchmark conditions:

$$\frac{1}{|S(T)|} \sum_{t \in S(T)} \text{Pass}(\mathcal{A}_{\text{base}}, t) \leq 0.05; \quad \forall \mathcal{A} \in \mathbb{A}_{\text{SOTA}}, \frac{1}{|S'(T)|} \sum_{t \in S'(T)} \text{Pass}(\mathcal{A}, t) \leq 0.10 \quad (2)$$

Condition (i) ensures that templates are not trivially solvable by a typical baseline, while condition (ii) ensures robustness across models, preventing leakage through model-specific weaknesses.

To ensure lasting reproducibility, all templates are released as open source with comprehensive documentation. Additionally, we commit to a one-year active maintenance through Oct.2026, including quarterly new SOTA model evaluations and empowering community contributions via open-sourcing. Details of our maintenance protocol and timeline are provided in Appendix D.3.

## 4.2 AGENTIC TASK SYSTEM

To generate dynamic, contextualized task instances while preserving *Collaboration-Necessary* properties, we develop an agentic task system around a GPT-4.1-powered agent (OpenAI, 2025). The agent instantiates static templates into realistic tasks by orchestrating structured tool invocations. Each instance is constructed from four components, generated by specialized tools:

➠ **TechnicalParameterTool**: Generates logic-critical parameters (e.g., data scales, performance thresholds) using rule-based methods constrained within pre-calibrated ranges.

➠ **ImplementationConstraintTool**: Selects framework versions and environment configurations from a validated list to define implementation boundaries.

➠ **ContextualVariableTool**: Produces realistic scenarios (e.g., company profiles, user roles, industry contexts) through constrained prompt generation.

➠ **InterfaceSpecificationTool**: Dynamically generates low-level interface elements such as function names, API endpoints, and variable naming conventions requiring code adaptation.

To ensure fairness and consistency across all rendered tasks, the agent strictly separates logic-critical generation performed by the **TechnicalParameterTool** and **ImplementationConstraintTool** from cosmetic and contextual generation, handled by the **ContextualVariableTool** and **InterfaceSpecificationTool**. Upon receiving a template, the agent analyzes its complexity indicators, such as domain requirements and algorithmic dependencies, and determines an optimal tool invocation sequence. Technical parameters are resolved first; if interdependencies arise, tools are coordinated iteratively using intermediate outputs as constraints. An internal error-checking mechanism monitors for parameter conflicts and triggers corrective re-invocation as needed. Once core parameters are fixed, the agent wraps the task in a realistic scenario, embedding surface-level ambiguity that requires requirement analysis to correctly interpret.

The agent produces two synchronized outputs: (1) a task instance that includes a README, starter code, project structure, and environment configuration for LLMs and humans; and (2) an evaluation script, accessible to the system, that extracts parameters and interface signatures to execute instance-specific checks. This structure enables consistent, automated evaluation pipelines per instance.

Finally, to ensure task instances are fair and benchmark-consistent, we validate agent output through a formal review process involving two independent human experts. By design, logic-critical components are produced deterministically within validated ranges, while contextual variables are generated under constrained prompts. This guarantees that variation does not introduce additional cognitive or technical difficulty. Expert evaluations achieved high inter-rater agreement (IRR), as detailed in Appendix D. Agent hyperparameters and prompt configurations are provided in Appendix C.

## 4.3 STANDARDIZED CLOUD IDE FOR HUMAN EVALUATION

Rather than building a custom development environment, HAI-Eval adopts an "outsourcing" philosophy by orchestrating a high-fidelity, industry-standard development workflow using established cloud tools. This approach offers a level of realism and capability far beyond the limited web editors used in traditional coding evaluations.

We leverage GitHub Codespaces (GitHub, Inc., 2021) to provide a standardized, fully-featured cloud IDE. Users access the test environment via their GitHub accounts. For each task instance, a new Codespaces environment is provisioned using a `devcontainer` file defined in the associated problem template. This file configures the remote VS Code instance and specifies the operating

system, runtime dependencies, and required extensions, including the critical Copilot extension. As one of the most widely used coding agents, Copilot has over 20 million users and is used by 90% of Fortune 100 companies (StackOverflow, 2024; TechCrunch, 2025). It supports multiple series of mainstream model backends, including GPT, Claude, and Gemini, and plays a key role in ecological validity by reflecting realistic developer workflows. When a user begins a task, Codespaces automatically launches a browser-accessible, preconfigured VS Code instance. This instance includes an integrated terminal, full file system access, a built-in debugger, and native Git version control support. By aligning with modern cloud development practices and eliminating confounding factors such as IDE familiarity, this setup ensures that performance reflects actual development skill. Each task includes a set of visible unit tests to assist users during implementation. Final scoring, however, is determined by a comprehensive suite of hidden test cases executed on the backend.

This workflow extends seamlessly into the evaluation process. Upon completing a task, the user executes a provided shell script in the terminal, which automatically packages the project directory and submits it via HTTPS POST to our backend endpoint. This endpoint, a robust and lightweight API implemented with FastAPI and deployed on a secure cloud server, validates the request, receives the submission, and forwards it to the evaluation system. The corresponding evaluation scripts are then executed, and a structured JSON file is returned containing a detailed performance breakdown. Evaluation metrics are formally defined in Section 5.1.

### 4.4 EVALUATION TOOLKIT FOR LLMS

To enable systematic and reproducible benchmarking of LLMs in realistic development settings, and to support comparative analysis between coding agents and human-AI teams, HAI-Eval includes an autonomous evaluation toolkit. The toolkit is built on top of VS Code and Copilot, allowing it to interact with any LLM supported by Copilot Agent mode. This design ensures ecological validity by aligning the evaluation environment with modern industry-standard workflows. We implement a dedicated VS Code extension, **HAI-Eval Controller (HAI-EC)**. HAI-EC is designed to replicate the **mechanical, procedural interactions** of a human developer using HAI-Eval, enabling direct comparability between LLMs and humans without introducing confounding variables. HAI-EC replaces Codespaces with Docker (Merkel, 2014) to locally deploy environments, improving efficiency while maintaining configuration fidelity. It iterates through the full task set, sequentially deploying isolated environments and orchestrating the evaluation workflow through a strictly defined automated pipeline for each task instance: (i) building the containerized environment; (ii) invoking Copilot via the VS Code API and uploading the task README to Copilot; (iii) triggering code generation and iteratively refining the solution by feeding back test execution logs; and (iv) once all visible tests pass or a predefined time limit is reached, executing the evaluation script, recording the results, and cleaning up the environment. This automated pipeline ensures consistent evaluation of coding agents across all tasks in HAI-Eval.

To ensure reproducible results, the evaluation toolkit uses a static dataset of 450 task instances rather than real-time dynamic task generation, in contrast to the human-facing evaluation setup. These instances are instantiated from 45 problem templates, with 10 unique tasks per template. All instances have been manually reviewed and validated to ensure consistency in quality and difficulty, as detailed in Appendix D. To support replication and benchmarking across different coding agents, the dataset is released as a standalone resource. This release strategy ensures that future researchers can benchmark new models against the same, consistent task distribution used in this work.

### 4.5 METRICS & EVALUATION SYSTEM

The HAI-Eval evaluation system uses a two-stage protocol to assess performance for any human or LLM. It first records five raw metrics per trial, then derives four aggregated metrics for analysis. A trial is defined as a single task completed by either a human developer or an LLM. For each trial, the system records the following raw metrics: (i) binary pass/fail outcomes for each functional test case; (ii) solution execution time; (iii) peak memory usage; (iv) total completion time; and (v) token usage from LLM interactions. To ensure fair and comparable evaluation of execution time and memory usage, we introduce **Auto-Calibrated Baselines**. Prior to each evaluation run, the system executes canonical reference solutions (representing efficient and inefficient implementations) to establish dynamic thresholds. These thresholds are used to convert continuous performance data into discrete binary pass/fail outcomes. This calibration process mitigates the effects of large-scale numerical variance across tasks and ensures platform-independent evaluation by adapting thresholds to the

current computational environment. The resulting binary efficiency metrics are robust, interpretable, and directly comparable across heterogeneous task instances.

We then derive four aggregated metrics that capture two core dimensions: solution quality and development efficiency. For solution quality: (i) **Overall Pass** (0/1) indicates whether the solution passes all test cases, including both functional and efficiency checks; (ii) **Partial Pass** (0-1) measures the proportion of test cases passed. For development efficiency: (iii) **Completion Time** records the total task-solving duration with a 60-minute timeout. Trials that fail or exceed the time limit are assigned a value of 60 minutes, following the standard penalized average runtime (PAR) approach; (iv) **Token Usage** captures the total number of used tokens.

## 5 EXPERIMENTS & ANALYSIS

We conduct a controlled user study alongside fully automated LLM evaluations. Our experiments are designed to measure performance under four distinct conditions that vary in the level of human–AI interaction, enabling us to measure human-AI synergy.

### 5.1 SETUP

**Experimental Conditions.**    We design 4 conditions of varying levels of human intervention:

➠ $C_H$ **(Human-Only):** Participants solve tasks independently without any AI assistance.

➠ $C_0$ **(Fully Autonomous AI):** Copilot operates without any human input, relying solely on its built-in robustness features (e.g., automatic retries, up to 25 attempts)

➠ $C_1$ **(Minimally-Intervened AI):** A researcher intervenes only in strictly defined procedural failures, with no logical or semantic assistance. Details are provided in Appendix F.3.

➠ $C_2$ **(Human-AI Collaboration):** Human developers freely use Copilot throughout the task.

$C_0$ and $C_1$ are evaluated using the static 450-instance dataset and HAI-EC. $C_0$ measures end-to-end autonomous agent performance, while $C_1$ isolates core reasoning capabilities by mitigating procedural execution failures. $C_H$ and $C_2$ are tested via our cloud IDE and dynamic task instances. This design enables direct comparison of human and AI contributions, and the measurement of synergy: human benefit ($C_2$ vs. $C_H$), AI benefit ($C_2$ vs. $C_0/C_1$), and failure isolation ($C_1$ vs. $C_0$).

**Human Study Design.**    To maximize statistical power, We conduct a within-subject, fully counterbalanced study where 45 participants each complete four tasks: two under $C_H$ and two under $C_2$. While order effects are a known concern in within-subject designs, we mitigate this risk through a combination of participant expertise and full counterbalancing. Our participants are expert users (highly proficient in programming, VS Code, and AI coding assistants) and prior research shows that such users are minimally influenced by task order (MacKenzie, 2002; 2024). To further control for order effects, we apply full counterbalancing along two dimensions. Each participant's task sequence is randomly selected from all balanced permutations of the four conditions. A Latin Square design ensures that every problem appears equally across conditions and is completed by different participants. The study is conducted over two sessions with a 24-hour interval to reduce cognitive fatigue. All participants complete their tasks via assigned anonymous GitHub accounts after a brief standardized training session. To ensure data quality, we record operational logs and perform manual spot checks to verify protocol adherence.

**Participants.**    We recruit 45 participants through personal contacts and advertisements posted on university forums. All participants are East-Asian current students or recent graduates (undergraduate to PhD) in Computer Science or related majors, and all regularly use coding agents. Participants were assigned to one of three professional tracks based on a detailed questionnaire assessing their background and technical skills. This academic demographic is strategically selected for its accessibility and relevance: (i) they frequently engage in unaided coding scenarios (e.g., exams, LeetCode), making them well-suited for $C_H$; and (ii) the pool is large enough to reliably screen for expert users. While this group is highly relevant, the generalizability to industry professionals and other ethnic groups is limited, so some care needs to be taken when drawing conclusions from our research. Full selection criteria and background details are provided in Appendices J and H.

**Technical Specifications.**    We evaluate all SOTA models supported by Copilot Agent mode as of July 26, 2025. Since Copilot does not permit hyperparameter customization, all models are evaluated with default settings. We select Claude-Sonnet-4 (Anthropic, 2025) as the ecologically

valid baseline across all conditions due to its market prevalence (Menlo Ventures, 2025). To assess model boundaries in $C_0$ and $C_1$, we additionally benchmark GPT-4.1, GPT-4o (Hurst et al., 2024), Claude-Sonnet-3.7 (Anthropic, 2025a), and Gemini-2.5-Pro (Comanici et al., 2025). We apply the four aggregated metrics defined in Section 4.5 and report both *pass@1* and *pass@10* in $C_0$ and $C_1$ to evaluate robustness. Statistical comparisons use appropriate tests with averaged results.

## 5.2 EXPERIMENTAL RESULTS

Our empirical results highlight a fundamental insight into the nature of human-AI synergy: the dynamic is shifting from a traditional human-as-tool-user paradigm to an emergent co-reasoning partnership, in which strategic breakthroughs can originate from either the human or the AI. Our Observations are presented as below.

Table 1: SOTA LLMs universally struggle on HAI-Eval, exposing a fundamental gap in reasoning.

| | Claude-Sonnet-4 | | Claude-Sonnet-3.7 | | GPT-4.1 | | GPT-4o | | Gemini-2.5-Pro | |
|---|---|---|---|---|---|---|---|---|---|---|
| | $C_0$ | $C_{1\Delta(\%)}$ | $C_0$ | $C_{1\Delta(\%)}$ | $C_0$ | $C_{1\Delta(\%)}$ | $C_0$ | $C_{1\Delta(\%)}$ | $C_0$ | $C_{1\Delta(\%)}$ |
| **Overall Pass@1 (%)** | 0.67 | $2.89_{\uparrow 2.22}$ | 0.00 | $1.56_{\uparrow 1.56}$ | 0.00 | $1.78_{\uparrow 1.78}$ | 0.00 | $0.00_-$ | 0.22 | $2.22_{\uparrow 2.00}$ |
| **Overall Pass@10 (%)** | 3.11 | $4.22_{\uparrow 1.11}$ | 0.44 | $2.40_{\uparrow 1.96}$ | 1.33 | $3.56_{\uparrow 2.23}$ | 0.00 | $0.00_-$ | 0.67 | $2.22_{\uparrow 1.55}$ |
| **Partial Pass@1 (%)** | 19.24 | $30.13_{\uparrow 10.89}$ | 8.71 | $17.47_{\uparrow 8.76}$ | 11.16 | $23.64_{\uparrow 12.48}$ | 5.82 | $12.09_{\uparrow 6.27}$ | 8.27 | $21.33_{\uparrow 13.06}$ |
| **Partial Pass@10 (%)** | 15.89 | $28.71_{\uparrow 12.82}$ | 12.05 | $20.34_{\uparrow 8.29}$ | 13.97 | $24.82_{\uparrow 10.85}$ | 7.63 | $15.28_{\uparrow 7.65}$ | 10.41 | $23.15_{\uparrow 12.74}$ |

***Obs* 1: SOTA LLMs hit a wall of higher-order reasoning.** Our experiments show that HAI-Eval's collaboration-necessary tasks pose a significant challenge to current SOTA LLMs. As shown in Table 1, all tested LLMs achieve near-zero pass rates under both $C_0$ and $C_1$, with the best-performing model achieving only an overall pass@10 of 4.22%. This result exposes *a core limitation of current LLMs: they are unable to perform higher-order reasoning tasks, such as interpreting complex requirements and planning multi-step solutions, revealing a performance gap that current LLMs cannot bridge alone*.

Table 2: Performance comparison of 4 conditions across difficulties. The final row shows the Averaged Overall Pass@1 across 3 difficulties.

| Easy | $C_H$ | $C_0$ | $C_1$ | $C_2$ |
|---|---|---|---|---|
| **Overall Pass@1 (%)** | 36.67 | 1.33 | 4.00 | **43.33** |
| **Partial Pass@1 (%)** | 52.50 | 27.60 | 43.20 | **62.90** |
| **Completion Time (s)** | 2165 | 127 | 142 | **1379** |
| **Tokens (M)** | 0 | 0.42 | 0.44 | **2.04** |

| Medium | $C_H$ | $C_0$ | $C_1$ | $C_2$ |
|---|---|---|---|---|
| **Overall Pass@1 (%)** | 13.33 | 0.67 | 2.67 | **26.67** |
| **Partial Pass@1 (%)** | 27.50 | 18.53 | 29.20 | **50.80** |
| **Completion Time (s)** | 2889 | 216 | 235 | **2377** |
| **Tokens (M)** | 0 | 0.49 | 0.50 | **2.26** |

| Hard | $C_H$ | $C_0$ | $C_1$ | $C_2$ |
|---|---|---|---|---|
| **Overall Pass@1 (%)** | 6.67 | 0.00 | 2.00 | **23.33** |
| **Partial Pass@1 (%)** | 20.60 | 11.60 | 18.00 | **37.10** |
| **Completion Time (s)** | 3306 | 309 | 328 | **2814** |
| **Tokens (M)** | 0 | 0.58 | 0.62 | **2.31** |

| **Avg. Overall P@1** | 18.89 | 0.67 | 2.89 | **31.11** |

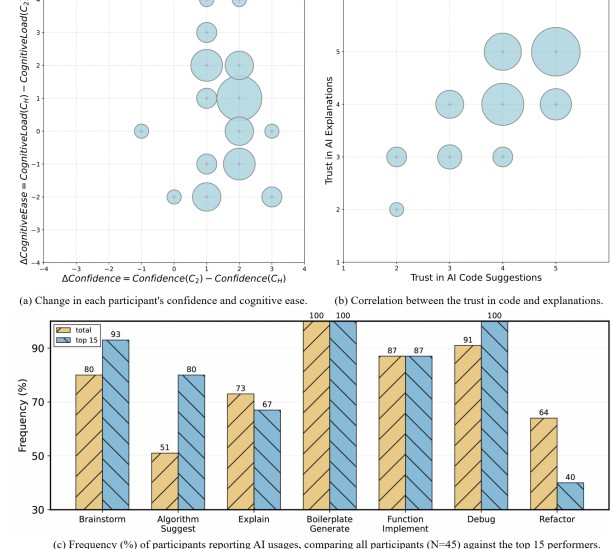

(a) Change in each participant's confidence and cognitive ease.

(b) Correlation between the trust in code and explanations.

(c) Frequency (%) of participants reporting AI usages, comparing all participants (N=45) against the top 15 performers.

Figure 4: Visualization of key participant feedback. Details of feedback statistics are provided in Appendix K.3.

***Obs* 2: The synergy paradigm can help address the gap.** In contrast to the limitations of standalone LLMs, pairing human users with coding agents yields substantial performance improvement. As shown in Table 2, the overall pass@1 rises to 31.11% under $C_2$, a statistically significant improvement over the 18.89% overall pass@1 achieved by unaided users ($C_H$), and far exceeding the best standalone LLM. This synergy also improves efficiency: average completion time under $C_2$ is significantly lower than in $C_H$, reflecting enhanced problem-solving fluency. This advantage is most pronounced on the most challenging tasks. Difficulty-wise analysis reveals that while unaided

human performance degrades sharply with increasing task difficulty, standalone LLM performance remains uniformly low. In contrast, collaborative performance remains stable across difficulty levels. This suggests that human problem-solving is constrained by the task's intrinsic strategic complexity, whereas LLMs are bottlenecked by a fundamental inability to parse contextual complexity. By bridging these complementary weaknesses, human-AI collaboration acts as a capability multiplier, especially on complex tasks. Together, these results provide the first strong quantitative evidence that *effective human-AI collaboration not only boosts productivity, but also enables the successful resolution of tasks that neither humans nor LLMs can solve alone*.

*Obs* 3: **LLMs are no longer execution tools, but co-reasoning partners.** To better understand the nature of this synergy, we analyze participant feedback and behavioral logs. The observations reveal a collaboration dynamic far more complex than the traditional user–tool model, in which humans lead and delegate execution tasks to AI. As shown in Figure 4.a, a plot of participants' confidence versus cognitive load reveals a prominent "cognitive shift": most participants reported a substantial increase in confidence without a proportional decrease in perceived mental effort. Critically, Figure 4.b shows a strong positive correlation between participants' trust in the AI's code and its explanations, suggesting they trusted the model's reasoning process as much as its outputs. This shift is further reflected in participants' self-reported usage patterns (Figure 4.c): while implementation support (e.g., generating boilerplate, debugging) was universally adopted, 80% of participants also used the AI for strategic brainstorming. Notably, 51% adopted a fundamentally different approach (e.g, algorithmic core, key library) proposed by AI. Among top performers, this strategic reliance on AI was particularly salient, where 12 of the 15 highest-performing participants reported leveraging this specific high-level capability. Expert analysis of user logs corroborates these self-reports, confirming a consistent pattern: high-performers engaged in active co-reasoning with the AI to discover and implement more effective solution paths. Together, these observations suggest that *LLMs are no longer mere execution tools, they are emerging as true co-reasoning partners*.

This evidence of a dynamic, co-reasoning partnership extends beyond the scope of prior studies. It emerges directly from the HAI-Eval framework, whose problem design is tailored not only to measure productivity, but to isolate and quantify the collaborative value of human–AI problem-solving. Additional analyses and track-specified results are provided in Appendix K.1 & K.2. We also provide a case study in Appendix K.3 to concretely illustrate this co-reasoning partnership through a fundamental approach shift.

## 5.3 HUMAN FEEDBACK ON TASK REALISM AND EVALUATION ACCURACY

Based on our post-test questionnaire in Appendix J.3, we collect participant feedback on task design and the fairness of our evaluation metrics. As shown in Table 3, the high mean scores across all items indicate a positive reception. Participants rate the tasks as highly realistic and consistently agree that our metrics accurately reflect their performance and effort in both $C_H$ and $C_2$. This feedback validates the authenticity of our problem space and the fairness of our evaluation, including the reasonableness of its hidden test cases. More human feedback results are provided in Appendix K.3.

Table 3: Summarized results of participant ratings on task realism and evaluation accuracy. The ratings use a 5-point Likert scale (1 = Strongly Disagree, 5 = Strongly Agree). SD indicates standard deviation.

| Statement | Mean | SD |
|---|---|---|
| Tasks reflected real-world challenges | 4.07 | 0.86 |
| *Accuracy of metrics in $C_H$:* | | |
|     Functional Correctness | 4.27 | 0.78 |
|     Efficiency Metrics | 3.82 | 0.88 |
| *Accuracy of metrics in $C_2$:* | | |
|     Functional Correctness | 4.20 | 0.85 |
|     Interaction Cost | 3.96 | 0.99 |

## 6 CONCLUSION

We introduce HAI-Eval, the first unified benchmark for quantifying human value in AI-assisted coding and challenging coding agents with tasks that necessitate human-AI collaboration. HAI-Eval reveals fundamental capability gaps between current LLMs and human developers in higher-order reasoning. We believe that HAI-Eval paves the way for defining developer competencies in the AI era and developing truly autonomous coding agents.

## ETHICS STATEMENT

For the experiments with participants, we strictly adhere to all ethical guidelines. All participants were clearly informed of the research purpose, procedures, potential task difficulty, and data usage, and signed informed consent forms allowing them to withdraw at any time. To protect privacy, all data has been anonymized, and we provided fair compensation for the participants. All procedures in this study were reviewed and approved by the Institutional Review Board (IRB) of the primary contributors' university.

As a benchmark including "human contribution" measurement, we recognize the potential for misuse. We stress that HAI-Eval is developed as a research tool to understand human-AI collaboration and the limitation of current SOTA LLMs. Its direct application in high-stakes evaluations, such as employee recruitment or performance reviews, should be approached with extreme caution, as it may introduce bias and inadequately capture the diverse spectrum of engineering talent.

## REPRODUCIBILITY STATEMENT

To ensure the reproducibility of our research and maximize its community value, we adopt a two-way open strategy.

First, to allow reviewers and the community to personally experience the core challenge of our benchmark, we provide a public, anonymous, and interactive demo. The goal is to provide an intuitive, first-hand understanding of the pre-specification task design that creates a significant performance gap for both standalone coding agents and unaided human developers, which can only be bridged by effective human-AI collaboration. This demo enables anyone to act as a participant, tackling "necessary collaboration" tasks within our environment with "ecological validity". We also provide the instruction for participants in our repository and a brief walkthrough in Appendix J.4.

Second, we provide the code and dataset for automatically evaluating LLMs on HAI-Eval, which allow any researcher to easily reproduce our results, more importantly, to use HAI-Eval to evaluate their own or future code models. Our goal is to make HAI-Eval a continuously evolving platform that serves the entire community, driving research into next-generation code agents capable of handling complex, real-world scenarios.

Our goal is to make HAI-Eval a continuously evolving platform that serves the entire community, simultaneously driving the development of next-generation code agents while also helping to define and cultivate the necessary skills for human developers in the AI era.

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

APPENDIX

## A    LIMITATION & FUTURE WORK

**Limitations.**    While our study provides a pioneering framework and valuable insights, we acknowledge several limitations. First, although HAI-Eval is theoretically language-agnostic, our current implementation is exclusively focused on Python. Second, our reliance on GitHub Copilot as a standardized interface, while beneficial for controlling experimental variables, constrained our evaluation to models currently supported by Copilot's agent mode. Consequently, several advanced models including o3 (OpenAI, 2025d) and GPT-5 (OpenAI, 2025), and prominent open-source models such as Deepseek (DeepSeek-AI, 2024; DeepSeek-AI et al., 2025), Llama (Dubey et al., 2024; Meta AI, 2025), and Qwen (Team, 2024; Yang et al., 2024; Hui et al., 2024; Yang et al., 2025) series were not included in our benchmark. Third, our user study participants consist entirely of East-Asian university students and recent graduates, all regularly using coding agents. While this group is highly relevant, the generalizability may be limited. Finally, to facilitate cross-task and platform-independent comparisons, we converted raw performance data into discrete pass/fail checks for our final analysis. While this approach is common practice on competitive programming platforms, it can impact the transparency and interpretability of our results.

**Future Work.**    Our future work will aim to address these limitations. Planned directions include: (i) extending the benchmark to additional programming languages (e.g., C++, Java) and new professional tracks (e.g., DevOps, Cybersecurity); (ii) broadening LLM evaluations to include models accessible via direct APIs; (iii) conducting follow-up user studies with experienced industry developers and participants from more ethnic diverse backgrounds; (iv) developing a more granular evaluation framework that reports detailed scores for correctness and various efficiency metrics separately. More substantially, we plan to enhance HAI-Eval by integrating emerging LLM-based review frameworks. This will enable us to move beyond purely quantitative metrics towards qualitative assessments of generated code, capturing crucial software engineering attributes such as readability, maintainability, and extensibility. Such advancements would further strengthen HAI-Eval as a holistic benchmark for both developer and AI capabilities.

## B    DETAILS OF PROBLEM TEMPLATE VALIDATION

**Model Selection.**    As shown in Figure 3, we use Claude-Sonnet-4 as the test model for the fidelity check, which is identical to the model used by Copilot in our human study. For the generalization test, we select two additional SOTA LLMs: GPT-4.1 and Gemini-2.5-Pro. Together, these three selected models represent the latest and most capable offerings of OpenAI, Anthropic, and Google, the leading providers of LLM services (Menlo Ventures, 2025), that are available in Copilot's Agent Mode. This selection ensures that our validation protocol reflects both the strongest commercially available models and the practical constraints of widely deployed developer tools.

**Validation Setup.**    Validation tests are conducted in an environment identical to that described in Section 5.1 for LLM evaluation. The entire testing process is under the $C_0$ condition, ensuring a standardized, objective, and reproducible assessment. Following Equation 2, in the fidelity check, a maximum of 1 out of the 20 instances for each template is permitted to pass, corresponding to a maximum pass rate of 0.05. In the generalization test, a maximum of one out of the 10 instances per template is allowed to pass, resulting in a maximum pass rate of 0.1.

**Detailed Results.**    Table 4 presents the validation results for our final template bank across three models. All 45 templates successfully meet both criteria in Equation 2. Notably, 42 templates achieve 0% pass rate across all models, demonstrating the effectiveness of our Collaboration-Necessary design principles.

Table 4: Validation results across models on final template bank.

| Metric | Claude-Sonnet-4 | GPT-4.1 | Gemini-2.5-Pro |
|---|---|---|---|
| Overall Pass Rate | 0.33% | 0% | 0.22% |
| Templates with 0% pass | 42 | 45 | 44 |

## C    AGENT CONFIGURATION AND PROMPT EXAMPLES

To ensure the transparency of our task instantiation process, this appendix provides comprehensive technical details of the task system. We describe the key hyperparameters used by the GPT-4.1

model that powers our agent, and we include representative prompt templates used to guide the agent in instantiation.

## C.1 HYPERPARAMETERS

The LLM in our task system, GPT-4.1, is configured with the hyperparameters below. This setup, particularly the use of a moderate temperature, ensures that task instances exhibit sufficient variability to prevent repetition, while maintaining the consistency necessary for controlled benchmarking.

- **temperature**: 0.7
- **top_p**: 0.9
- **max_tokens**: 8192

## C.2 PLANNING PROMPT EXAMPLE

---

**Prompt for Task Planning**

You are a task agent. Analyze the given template {template_info} and then:

1. Based on the potential tool list provided by the template, determine which tools are needed

2. According to the tool dependency information provided in {tool_info}, plan the invocation sequence

3. Based on the scenario and value ranges provided by the template, set appropriate parameters for each tool

You can:

- Adjust tool invocation order based on template characteristics

- Skip unnecessary tools

- Repeatedly invoke the same tool for parameter refinement

Below are two examples. Please return the execution plan following this JSON format.
**Example 1 (Network Optimization Problem):**

```
{
  "template_id": "SDE-HARD-001",
  "difficulty": "hard",
  "track": "SDE",
  "execution_plan": [
    {
      "step": 1,
      "tool": "TechnicalParameterTool",
      "parameters": {
        "num_nodes_range": [10, 30],
        "edge_ratio_range": [1.2, 2.5],
        "budget_range": [3, 10]
      },
      "rationale": "First generate network scale parameters
                    as foundation for all subsequent tools"
    },
    {
      "step": 2,
      "tool": "ImplementationConstraintTool",
      "parameters": {
        "required_libraries": ["networkx", "json"],
        "python_version": "3.8+"
      },
      "dependencies": ["step_1_output"],
```

```
      "rationale": "Determine tech stack based on network scale"
    },
    ...
  ]
}
```

## C.3 EXECUTION PROMPTS EXAMPLE

**Prompt for Task Execution**

You are now executing the plan. You will receive:

1. The execution plan (execution_plan)

2. Tool interface specifications (tool_interfaces)

3. Current step information (current_step)

Your tasks are:

1. Execute strictly in the step order specified in the plan

2. Invoke tools using parameters specified in the plan

3. Use outputs from previous steps as dependency inputs for current step

4. Handle potential errors from tool invocations

If tool invocation fails, you should:

- Analyze the failure reason

- Adjust parameters and retry (maximum 3 attempts)

- If failures persist, mark the step as failed with explanation

Please return execution results for each step in the following format:

**Execution Example:**

```
{
  "step": 1,
  "tool": "TechnicalParameterTool",
  "status": "success",
  "execution": {
    "attempt": 1,
    "input_parameters": {
      "num_nodes_range": [10, 30],
      "edge_ratio_range": [1.2, 2.5]
    },
    "tool_response": {
      "num_nodes": 23,
      "num_edges": 42,
      "upgrade_budget": 7
    }
  },
  "validation": {
    "is_valid": true,
    "checks": ["Parameters within range",
               "Edge-node ratio reasonable"]
  }
}
```

## D TASK INSTANCE VALIDATION AND QUALITY CONTROL

To ensure the integrity, fairness, and consistency of task instances within the final dataset for LLM evaluation, we implement a comprehensive quality control protocol that verifies task validity and

adherence to the "Collaboration-Necessary" design principles. This section details the validation methodology, review procedures, and quality control metrics. All validation work was conducted independently of the human study by two domain experts, each with over three years of industry experience in software engineering and AI, or equivalent academic research credentials.

## D.1 VALIDATION METHODOLOGY

Our quality control protocol comprises two complementary validation methods: (1) Document Review, and (2) Manual Testing. In **Document Review**, experts systematically examine each task instance, including task descriptions, code frameworks, configuration files, and other supporting materials, to assess structural soundness, clarity, and internal consistency. This method is applied to all task instances. In **Manual Testing**, experts attempt to implement a subset of task instances to verify solvability and confirm the presence of *Collaboration-Necessary* characteristics. Due to resource constraints, this method is applied using a representative sampling strategy.

To ensure both efficiency and rigor in the evalidation process, we design a two-stage review protocol:

➡ *Initial Validation.* For each of the 45 problem templates, we use the agentic task system (Section 4.2) to generate an initial set of **3** task instances. Two independent experts conduct validation on each instance validate each instance through both document review and manual testing. All three instances must pass both validation stages to proceed. If any instance fails, the template is flagged for revision, and adjustments are made either to the template itself or the task system's tools based on the identified failure. Revised templates must restart the full validation process from the first stage.

➡ *Extended Validation.* Templates that successfully pass the first stage to a second round, where the task system generates an additional **7** task instances. The same two experts apply identical validation criteria. At this stage, document review is performed on all seven instances, while manual testing is conducted on two randomly selected instances. All instances must pass their respective validation checks. Any failure triggers a return to the revision process, after which validation must restart from the first stage. Only templates that successfully complete both stages contribute their full set of **10** validated task instances to the static evaluation dataset.

This process not only validates the effectiveness of individual templates from a performance perspective, but also verifies the reliability of both the dynamic task system and the resulting static evaluation dataset. Through this rigorous quality control pipeline, we ensure that the task system consistently instantiates tasks that meet benchmark standards, while guaranteeing that each task included in the final static dataset satisfies our design requirements.

## D.2 DIFFICULTY CALIBRATION AND QUALITY CRITERIA

Our difficulty calibration and quality examination follows a rigorous two-stage process: (1) **Objective Indicator Assessment**, where tasks are measured against specific metrics; and (2) **Expert Calibration**, where domain experts validate these metrics against standards. Experts evaluate each task instance across four critical dimensions, issuing binary judgments (Pass/Fail). Notably, the Difficulty Consistency metrics are derived from established software engineering literature (Pelánek et al., 2022). A task instance is considered valid only if it passes all dimensions simultaneously:

- **Requirement Clarity:**
  - *Task Description Precision:* The task description provides an unambiguous specification of the objective.
  - *Context Completeness:* All necessary background information, assumptions, and constraints are clearly stated.
  - *Criteria Definition:* Evaluation metrics and expected outcomes are clearly defined.
- **Code Framework Correctness:**
  - *Initial Code Validity:* All provided starter code compiles and runs without errors.
  - *Dependency Completeness:* All required libraries and tools are properly declared.
  - *Environment Setup:* The functional development environment can be successfully instantiated with the provided configuration.
- **Difficulty Consistency:**
  - *Algorithmic Complexity:* The core algorithmic challenges align with the defined difficulty level and remain consistent across instances.

- *Implementation Scope:* The volume of required code and functional components falls within the pre-calibrated range for the assigned difficulty.
- *Time Requirements:* The estimated completion time for a qualified expert aligns with the designated difficulty category.

- **Collaboration-Necessary Characteristics:**

  - *Requirement Analysis Necessity:* Solving the task requires meaningful interpretation and decomposition of ambiguous or complex requirements.
  - *Contextual Reasoning:* The solution requires understanding of domain-specific constraints or business logic that cannot be directly inferred through pattern-matching.

### D.3    INTER-RATER AGREEMENT & PASS RATE ANALYSIS

**Inter-Rater Agreement.**    To ensure the objectivity and reliability of quality assessments, we calculated inter-rater agreement (IRR) between the two expert reviewers. IRR was computed on a subset of 90 task instances selected via stratified sampling, with exactly two instances sampled from each template. For each instance, both experts independently provided binary judgments (Pass/Fail) on each of the four quality dimensions.

We report IRR using Cohen's $\kappa$ (Cohen, 1960), calculated for each quality dimension. As shown in Table 5, all dimensions achieved "substantial agreement" ($\kappa > 0.80$) or higher (Landis & Koch, 1977). **Notably, the Difficulty Consistency metric achieved a score of 0.97**, demonstrating that our two-stage determination process yields highly stable and objective difficulty classifications. For cases where the two experts initially disagreed on any dimension during the initial assessment, we conducted structured review discussions and revision processes. All such instances were revisited and updated to ensure both experts agreed that the tasks fully satisfied all quality dimensions.

Table 5: Inter-rater agreement results across dimensions.

| Quality Dimension | Cohen's $\kappa$ |
|---|---|
| Requirement Clarity | 0.93 |
| Code Framework Correctness | 1.00 |
| Difficulty Consistency | 0.97 |
| "Collaboration-Necessary" Characteristics | 0.91 |

**Pass Rate.**    Among the 45 problem templates submitted for manual validation, 40 successfully passed both validation stages on their first attempt. Two templates required one round of revision after first-stage failures, one template required revision after a second-stage failure, and two templates were ultimately rejected and replaced. These results demonstrate that our task system consistently produces high-quality, structurally valid task instances from validated templates.

The 88.9% first-attempt pass rate reflects the system's ability to apply parameterized adjustments while preserving core task characteristics. Simultaneously, this validation protocol ensures that every task included in the static dataset meets benchmark design standards, exhibiting consistent difficulty levels, unambiguous specifications, and robust *Collaboration-Necessary* properties.

## E    MAINTENANCE AND REPOSITORY GOVERNANCE

HAI-Eval is designed as a benchmark to track the evolving capability frontier between state-of-the-art LLMs and human developers. Given the rapid pace of model improvements, a static benchmark would quickly become outdated, reducing its scientific value. To address this, we establish a maintenance protocol focused on community governance and reproducibility, to ensure that HAI-Eval remains both challenging and representative, continuously upholding its value as a *Collaboration-Necessary* benchmark.

### E.1    REPOSITORY GOVERNANCE

To ensure transparency, usability, and community alignment, we not only open-source HAI-Eval, but also establish clear governance practices for the repository of HAI-Eval. These practices govern how updates are versioned and how the community can participate in task refinement and expansion, supporting HAI-Eval as a sustainable and evolving benchmark.

**Versioning.**    All major updates to HAI-Eval are tracked using semantic versioning (e.g., v1.1, v2.0). Each version is accompanied by a detailed changelog published on the project's repository.

**Community Engagement.** We actively welcome community contributions through GitHub issues and requests, including reporting bugs, identifying quality concerns, or proposing maintenance for existing templates. Moreover, we encourage the community to propose new templates and *Collaboration-Necessary* designs, which can be vetted and incorporated into future community-led forks or versions.

**Open Leaderboard.** We maintain an open leaderboard in the Github repository of HAI-Eval to track most advanced model performance. Our maintenance plan, starting in Q4 2025, involves conducting quarterly evaluations at the end of each period. These evaluations will benchmark new SOTA models supported by Copilot Agent mode. For instance, based on the updated support after the submission of this paper, our next evaluation cycle will include Claude Sonnet 4.5 (Anthropic, 2025b), GPT-5 (OpenAI, 2025), GPT-5-Codex (OpenAI, 2025c), GPT-5.1 (OpenAI, 2025a), GPT-5.1-Codex (OpenAI, 2025b), and Grok Code Fast 1 (xAI, 2025). While our team provides these regular updates, we also strongly support community contributions. Researchers are encouraged to use our open-source toolkit to run evaluations on new models and submit their results via pull requests. Upon verification, these community-driven results will be integrated, ensuring it remains a comprehensive and up-to-date resource.

### E.2 MAINTENANCE PROCESS

To maintain the integrity and consistency of the benchmark, all contributions that modify or replace templates, whether from our team or the community, should adhere to the following maintenance process.

**Step 1: Re-validation of Existing Templates.** Any contribution that proposes replacing a template must demonstrate that the existing template is "compromised" by conducting *pass@1* tests on 10 dynamic tasks that show the template's *pass@1* score on a certain new SOTA model exceeds our defined generalization test threshold in Section 4.1 ($>0.10$).

**Step 2: Validation of New Templates.** New templates (including both replacement and expansion) must pass the full validation protocol. We encourage contributors to use our open-source toolkit to run the automated validation checks and include the results in their pull request. Our team will then conduct the final manual review to ensure the template meets the both *Ecological Validity* and *Collaboration-Necessary* design principles, such as requirement clarity and code correctness.

**Step 3: Synchronization of the Static Dataset.** Any update to a template must be accompanied by 10 new, manually-reviewed static instances for the evaluation toolkit. This ensures the static dataset always reflects the most up-to-date and representative task set for reproducible benchmarking.

## F EXAMPLES OF TASK INSTANCES ACROSS TRACKS & DIFFICULTY

To illustrate the diversity and complexity of tasks in HAI-Eval, we present representative examples across professional tracks and difficulty levels. The following subsections showcase sample instances from the Data Science (DS), Machine Learning Engineering (MLE), and Software Development Engineering (SDE) tracks at varying levels of difficulty.

### F.1 DS-HARD

**Task Description:** The task description below is a short summary of the original text. The original version is more verbose to make it hard for LLM parsing. This makes human-LLM cooperation necessary, a central idea in our framework.

> The primary objective is to develop a robust algorithm for calculating a proprietary metric, the **Prime Impact Aggregate,** across a rolling time window of customer transactions. Given a time-series list of `transaction_values`, a window size $k$, and a number of top segments $x$, the task is to compute this aggregate for every contiguous sub-array (cohort) of length $k$. The calculation for a single cohort begins with a frequency analysis of its transaction values. From this analysis, the top $x$ most frequent transaction values are identified as the "core segments." A tie-breaking rule specifies that if two values share the same frequency, the one with the higher value is prioritized. The **Prime Impact Aggregate** is then the sum of all occurrences of these identified core segment values within the cohort; all other values are ignored. A special case exists where if a cohort contains fewer than $x$ distinct transaction

values, the aggregate is simply the sum of all transactions in that cohort. The final deliverable is a list of integers, where the $i$-th element represents the calculated aggregate for the cohort starting at index $i$, resulting in an output list of length len(transaction_values) $- k + 1$.

**Folder Structure:**

- `README.md`: This is a complete, human-readable description of the problem.
- `solution.py`: This is the user's deliverable. It contains all starter code.
- `task_data.json`: This is an example test case given to the user. The true test dataset contains hidden testcases.

**Starter Code:**

```python
def calculate_rolling_cohort_sum(transactions, size, topk):
    """
    For each window of size `size`, this function finds the sum
    of values that belong to the `topk` most frequent categories.
    """
    # --- Your implementation starts here ---
    # [Student Code]
    # --- Your implementation ends here ---

if __name__ == '__main__':
    transactions, k, x = load_test_data("task_data.json")
    result = calculate_rolling_cohort_sum(transactions, k, x)
    print(f"Result preview: {result[:5]}")
```

**Key Challenge:** The key challenge is the efficient management of state across sliding windows. The difficulty lies in creating a system that can incrementally update the set of top segments as one transaction enters the window and another exits, without re-sorting or recounting the entire window each time.

**Evaluation Logic:**

*Correctness.* The participant's solution correctness is verified by comparing its output list against the one produced by a simple, brute-force implementation. The two lists must be identical.

*Performance.* The solution performance is measured by running it against a large dataset. A successful solution must be significantly faster than the naive approach to pass the evaluation.

## F.2 MLE-MEDIUM

**Task Description:** The task description below is a short summary of the original text. The original version is more verbose to make it hard for LLM parsing. This makes human-LLM cooperation necessary.

The primary objective is to detect the longest **palindromic signature** within a network security graph. Given a network with $n$ nodes (where each node has a security label character), edges representing bidirectional connections, and a string of security codes, the task is to find the maximum length palindromic sequence that can be formed by traversing the network. The traversal begins at any node and moves through adjacent nodes, collecting their security labels to form a signature. Each node can be visited at most once during a single traversal. A palindromic signature represents a symmetric security pattern that could indicate verified bidirectional communication channels or encrypted pathways. The algorithm must explore all possible paths through the network using depth-first search with backtracking, checking if the collected labels form a palindrome at each step. The final deliverable is an integer representing the length of the longest palindromic signature achievable through any valid traversal of the network graph.

**Folder Structure:**

- `README.md`: This is a complete, human-readable description of the problem.

- `solution.py`: This is the user's deliverable. It contains all starter code.

- `task_data.json`: This is an example test case given to the user. The true test dataset contains hidden testcases.

**Starter Code:**

```python
def find_longest_palindrome_path(n: int, edges: List[List[int]],
                                 label: str) -> int:
    """
    Finds the longest palindromic pattern in the network graph.
    """
    if n == 0:
        return 0

    # Build adjacency list
    graph = defaultdict(list)
    for u, v in edges:
        graph[u].append(v)
        graph[v].append(u)

    max_length = 1  # Single character is always a palindrome

    def is_palindrome(s: str) -> bool:
        """Check if a string is a palindrome."""
        return s == s[::-1]

    def dfs(node: int, visited: Set[int], path: str) -> None:
        """Depth-first search to explore all paths."""
        nonlocal max_length
        # --- Your implementation starts here ---
        # [Student Code]
        # --- Your implementation ends here ---

    # Try starting from each node
    for start_node in range(n):
        # TODO: Initialize DFS from each starting node
        pass

    return max_length

if __name__ == "__main__":
    task_data = load_task_data()
    result = find_longest_palindrome_path(task_data['n'],
                                          task_data['edges'],
                                          task_data['label'])
    print(f"Longest palindromic signature: {result}")
```

**Key Challenge:** The key challenge is efficiently exploring all possible paths through the graph while tracking visited nodes and checking for palindromes. The difficulty lies in implementing an optimized backtracking algorithm that can prune unnecessary branches early when the remaining unvisited nodes cannot possibly form a longer palindrome than the current maximum.

**Evaluation Logic:**

*Correctness.* The participant's solution correctness is verified by comparing its output against the expected maximum palindrome length for various test graphs. The solution must correctly handle edge cases including disconnected nodes, single-node graphs, and graphs with no palindromic paths longer than 1.

*Performance.* The solution performance is measured on large graphs with up to several hundred nodes. A successful solution must employ efficient pruning strategies to avoid exploring all $O(n!)$ possible paths, achieving reasonable runtime through techniques such as early termination and dynamic programming optimizations where applicable.

### F.3 SDE-EASY

**Task Description:** The task description below is a short summary of the original text. The original version is more verbose to make it hard for LLM parsing. This makes human-LLM cooperation necessary.

The primary objective is to implement a counting algorithm for **binary palindromic numbers** within a specified range. Given a non-negative integer $n$, the task is to count all integers $k$ where $0 \leq k \leq n$ such that the binary representation of $k$ (without leading zeros) reads the same forwards and backwards. A binary palindrome is defined as a number whose binary string representation is symmetric. For example, 5 (binary: 101) and 7 (binary: 111) are binary palindromes, while 6 (binary: 110) is not. The algorithm must efficiently handle large values of $n$ up to $10^9$. While a brute-force approach checking each number individually works for small inputs, an optimized solution should leverage the mathematical patterns of binary palindromes, potentially generating them directly rather than checking all numbers. The final deliverable is a single integer representing the total count of binary palindromic numbers in the inclusive range $[0, n]$.

**Folder Structure:**

- `README.md`: This is a complete, human-readable description of the problem.
- `solution.py`: This is the user's deliverable. It contains all starter code.
- `task_data.json`: This is an example test case given to the user. The true test dataset contains hidden testcases.

**Starter Code:**

```python
def count_binary_palindromes(n: int) -> int:
    """
    Counts the number of binary palindromic numbers from 0 to n.

    A binary palindrome is a number whose binary form reads the
    same forwards and backwards.
    """
    if n < 0:
        return 0

    count = 0

    def is_binary_palindrome(num: int) -> bool:
        """Check if a number's binary form is palindromic."""
        binary = bin(num)[2:]  # Remove '0b' prefix
        return binary == binary[::-1]

    # --- Your implementation starts here ---
    # [Student Code]
    # --- Your implementation ends here ---
```

```
21
22      return count
23
24  if __name__ == "__main__":
25      task_data = load_task_data()
26      n = task_data['n']
27      result = count_binary_palindromes(n)
28      print(f"Total binary palindromes found: {result}")
```

**Key Challenge:** The key challenge is developing an efficient algorithm that can handle large values of $n$ (up to $10^9$). While a brute-force approach checking each number is straightforward, it becomes computationally expensive for large inputs. The optimal solution requires understanding the mathematical patterns of binary palindromes and potentially generating them directly based on bit-length patterns rather than checking every number in the range.

**Evaluation Logic:**

*Correctness.* The correctness is verified by comparing the output count against the expected number of binary palindromes for various test cases. The solution must correctly handle edge cases including $n = 0$, small values where brute force is acceptable, and large values up to $10^9$.

*Performance.* The solution performance is measured on large inputs. A successful solution must complete within reasonable time limits for $n$ values up to $10^9$. Solutions that use brute-force checking for every number will likely timeout, while optimized approaches that leverage palindrome generation patterns or mathematical formulas should pass the performance requirements.

## G  STANDARDIZED INTERVENTION PROTOCOL FOR CONDITION $C_1$

The minimally human-intervened condition, denoted as $C_1$, is designed to measure the upper bound of a large language model's core logical reasoning capabilities by eliminating common non-logical, procedural obstacles. In this setting, researchers simulate an "idealized execution environment" and are strictly prohibited from intervening in the model's logical reasoning or problem-solving process in any form. Minimal assistance is permitted only in narrowly defined cases of procedural failure, where the model's output is correct in intent but fails due to environmental or infrastructural constraints. The following list outlines the only permitted procedural interventions under the $C_1$ condition. Each case reflects a well-scoped operational exception where intervention restores intended task execution without assisting with reasoning, decision-making, or problem decomposition.

- **Environment and Command Invocation Errors**
    - *Description:* The model generates the correct command or script but executes it in an incorrect runtime context (e.g., wrong Conda environment or virtual environment).
    - *Permitted Intervention Example:* If the model runs `python main.py` instead of the required `conda run -n myenv python main.py`, the researcher may intervene to correct the command.

- **Missing Dependency Errors**
    - *Description:* The code fails due to a missing library that was explicitly declared in a project dependency file (e.g. `requirements.txt`). This typically occurs when the environment fails to automatically install all declared dependencies. No intervention is allowed if the dependency was not declared by the model.
    - *Permitted Intervention Example:* If the code fails due to a missing library such as `pandas` and that library was listed in `requirements.txt`, the researcher may manually execute `pip install pandas` to install the intended library.

- **File/Directory Permission Issues**
    - *Description:* The generated code fails due to insufficient file or directory permissions, such as attempting to execute a non-executable script or write to a read-only location. These are considered environmental issues rather than logic errors.
    - *Permitted Intervention Example:* If the model generates a script without execution permissions, the researcher may run `chmod +x runscript.sh` to allow execution.

- **Port Conflicts or Basic Network Configuration Errors**
  - *Description:* In tasks involving network services, the model may attempt to start a service on a port that is unavailable due to environmental constraints (e.g., already in use or restricted). These errors are considered infrastructure-level and not part of the model's reasoning capabilities.
  - *Permitted Intervention Example:* If the model attempts to start a service on an occupied port such as `8000`, the researcher may modify the configuration to use an available port, such as `8001`.

- **Missing or Incorrect Environment Variables**
  - *Description:* The model correctly attempts to read a configuration value (e.g., an API key or file path) from an environment variable that is expected to be defined as part of the task setup, but the variable is missing or incorrectly set due to an environment configuration issue.
  - *Permitted Intervention Example:* If the model references `os.environ['DATA_DIR']` and this variable is unset, the researcher may execute `export DATA_DIR=/path/to/data`, provided the expected configuration is explicitly or implicitly required by the task.

## H  DEMOGRAPHIC STATISTICS OF PARTICIPANTS

### H.1  PARTICIPANT SELECTION CRITERIA

Participant credentials were verified via publicly available materials, including GitHub profiles, Google Scholar pages, and LinkedIn profiles. For selected participants, we also conducted brief online interviews in which candidates completed a designated task using VS Code with Copilot, allowing us to validate their practical proficiency in AI-assisted programming. Additional details regarding data collection and privacy safeguards are provided in Appendix I. To ensure professional competency with experimental tools and minimize learning effects related to tool unfamiliarity, all participants were required to meet the following eligibility criteria:

- At least 18 years of age
- Major in Computer science, software engineering, or related field
- Minimum of two years of programming experience and demonstrated proficiency in unassisted programming
- Proficient in using Visual Studio Code
- Regular use of AI programming assistants (e.g., Copilot, Claude Code) at least three times per week in the past month
- Proficient in Python programming
- Proficient in the technology stack relevant to their assigned track
- English reading and writing proficiency sufficient to understand task requirements

Based on participants' professional backgrounds and stated interests, we assigned them to one of three professional tracks: Software Development Engineer (SDE), Machine Learning Engineer (MLE), or Data Scientist (DS), with 15 participants in each track. Given the complexity of each individual's personal circumstances, these criteria were used as reference guidelines rather than strict eligibility rules for participant assignment. Track assignment criteria are as follows:

- **Academic background alignment**: Prior coursework in core computer science topics for SDE, machine learning or deep learning for MLE, and statistics or data analysis for DS.
- **Project experience type**: Experience with system development for SDE, model or algorithm development for MLE, and data processing or visualization for DS.
- **Career goal consistency**: Track preferences were aligned with participants' self-reported career plans and professional interests.

## H.2 DEMOGRAPHIC STATISTICS

All 45 participants identified as East Asian, including 28 males and 17 females. Ages ranged from 19-26 years (mean = 21.4). Educational backgrounds include 24 participants with or currently pursuing undergraduate degrees, 15 with or currently pursuing master's degrees, and 6 with or currently pursuing doctoral degrees. Participants reported an average of 1.47 internship experiences, with 84.4% using LLMs for coding assistance on a daily basis. Participants' current or previous academic affiliations span institutions across the United States, Mainland China, Hong Kong, and Singapore. Additional details are provided in Figure 5.

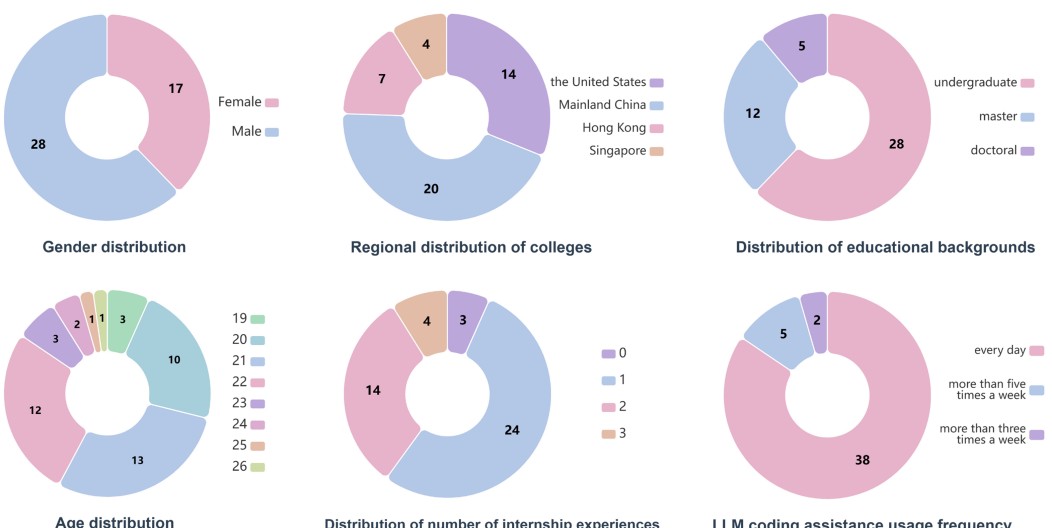

Figure 5: Visual representations of the participants' demographic data.

As discussed in Appendix A, we acknowledge that a limitation of this study is the demographic homogeneity, particularly in terms of race and cultural background. While this homogeneity may enhance internal validity by minimizing potential cultural confounds, it also limits the generalizability of our findings to the broader global developer population. In future work, we aim to replicate this study with participants from a wider range of cultural backgrounds to evaluate the cross-cultural robustness of the human-AI collaboration patterns observed in this study.

## I CONFIDENTIALITY STATEMENT

This section details the protocol we implemented to ensure the strict confidentiality of participant information throughout the study.

**Participant Anonymity during the Study.** The primary mechanism for ensuring participant anonymity during the experimental phase was the use of pre-assigned, anonymous GitHub accounts. Participants were explicitly instructed not to use their personal GitHub accounts. Instead, the research team created and provided a unique, anonymous GitHub account for each participant. The login credentials for these anonymous accounts were securely communicated to each participant via their registered email address prior to their first session. By using these pre-assigned accounts, we ensured that all actions within the experimental environment, including code submissions, IDE interactions, and operational logs, were decoupled from the participant's real-world identity from the outset.

**Participant Information Verification.** To ensure the integrity of our participant pool and the validity of self-reported expertise, we conducted a verification process based on the information provided in the pre-test questionnaire, detailed in Appendix J.2. The verification involved cross-referencing publicly available information based on participants' names and institutional affiliations. For example, we looked up academic profiles (e.g., publications on Google Scholar) and professional networking sites (e.g., work experience on LinkedIn) to validate self-reported credentials. This step was performed solely to validate that self-reported information met our selection criteria. All information accessed during verification was handled with strict confidentiality by the core research team and was used exclusively for eligibility screening.

**De-identification and Aggregation.** After data collection, a rigorous de-identification protocol was executed to ensure complete participant anonymity. During analysis, we used pre-assigned anonymous GitHub usernames as sole identifiers while permanently removing all personally identifiable information, including names, email addresses, and verification details, from the research dataset. The sensitive information was stored separately in encrypted files with restricted access for compensation only. The anonymized experimental data and questionnaire responses were linked using only the anonymous GitHub usernames. All published findings and any released data are presented in aggregated, fully anonymized format, making individual participant identification impossible.

## J    USER STUDY MATERIALS & DETAILS

This section presents all materials provided to the user study participants, including the informed consent form, the pre-test screening questionnaire, and the post-test feedback questionnaire.

### J.1    INFORMED CONSENT FORM

*The following is a static version of the informed consent form for inclusion in this appendix. All identifying information has been replaced with placeholders. In the actual study, participants were required to check a box and sign, indicating they had read, understood, and voluntarily agreed to participate.*

**Research Project Title:** HAI-EVAL: EVALUATE HUMAN VALUE IN AI-ASSISTED CODING
**Principal Investigator:** xxxxxxx
**Institution:** xxxxxxxxxxxxxx
**Contact Information:** xxxxxxxxxxxxxx@xxx.xxx

#### 1. INTRODUCTION

You are invited to participate in an academic study aimed to develop a novel framework for assessing programming abilities with coding agents. Your participation will provide valuable scientific data for understanding the core value of programmers in the AI era and for improving future engineering education and technical interviews. This test is conducted entirely in English and therefore requires proficiency in English reading and writing. To ensure the ethical conduct of this research and the protection of all participants, this study has been reviewed and approved by the Institutional Review Board (IRB). If you have any questions, please contact the Principal Investigator.

#### 2. PROCEDURES

If you agree to participate in this study, you will be asked to complete the following:

- **Pre-Test Questionnaire:** You will first complete an online questionnaire (approx. 10 minutes) to provide basic information, educational background, and technical experience. This helps us ensure you meet the study's criteria.

- **Programming Tasks & Conditions:** If you are selected as a participant, you will complete a total of four programming tasks related to your selected professional track in a pre-configured cloud IDE. The tracks include Software Development Engineer (SDE), Machine Learning Engineer (MLE), and Data Scientist (DS). The IDE is developed based on GitHub Codespaces and Visual Studio Code. The experiment follows a within-subject design, meaning you will experience both experimental conditions:
    - Two tasks will be completed in a **human-only condition**, without the aid of GitHub Copilot.
    - Two tasks will be completed in a **human-AI collaboration condition**, with GitHub Copilot enabled.
    - The order of tasks and conditions will be fully counterbalanced to prevent order effects.

- **Time Commitment and Arrangement:**
    - We estimate each task will take approximately 45-60 minutes. The total time commitment is expected to be around 3-4 hours.

- The experiment is divided into two sessions separated by a 24-hour interval to mitigate fatigue. You will complete two tasks per session. Ideally, the interval between the two sessions should be exactly 24 hours. However, if you have important personal business, please inform us via email. After reviewing your request, we may provide a flexible window of up to two hours.

- **Data Collection:** The system will automatically collect your final submitted code and performance metrics for evaluation. To guarantee data quality, we will also record operational logs.

- **Post-Test Feedback:** After finishing all tasks, you will be asked to complete a brief final questionnaire (approx. 5-10 minutes) to provide feedback on your experience, which should be completed within one day of finishing all tasks.

### 3. RISKS AND BENEFITS

- **Risks:** As approved by the IRB, the risks associated with this study are minimal. You may experience some stress or frustration. All your personal data will be strictly anonymized.

- **Benefits:** You will gain insight into a novel evaluation method.

### 4. COMPENSATION

Upon completion of all four programming tasks and the questionnaires, selected participants will be compensated with 40 USD or the equivalent amount in another currency. Payment will be made via one of the following methods: Amazon Gift Card, PayPal, Zelle, Alipay, or WeChat. The specific method will be determined in consultation with you after the study is complete.

### 5. CONFIDENTIALITY

We will take strict measures to protect your privacy.

- **Anonymous Access:** To ensure your anonymity, you will not use your personal GitHub account. You will be provided with a uniformly assigned, anonymous GitHub account to access Codespaces for the tasks. The credentials for this account will be sent to your registered email address.

- **Data Usage:** The personal information you provide will be used for specific, distinct purposes. Your contact information, typically email, will be used strictly for study-related communication and compensation. Your demographic and background information will be used for anonymized statistical analysis in our research.

- **Publication:** All published research findings will use fully anonymized, aggregated data. No information that could personally identify you will be disclosed.

### 6. VOLUNTARY PARTICIPATION

Your participation is voluntary. You may withdraw at any time without penalty.

### J.2 PRE-TEST QUESTIONNAIRE

*The following questionnaire was administered to screen and assign participants. For inclusion in this appendix, all interactive input fields have been removed.*

This questionnaire is designed to understand your background to ensure you meet the participation criteria for this study and to assign you to the most suitable task group. The information you provide will be kept strictly confidential. Please ensure that all information you provide is truthful and accurate. We reserve the right to withhold compensation if any of the information is found to be false or does not match the actual situation. We will try our best to arrange you to the role applied by you, but we do not guarantee it.

PART 1: BASIC INFORMATION & ROLE SELECTION

1. Name:

2. Email Address:

3. Which role are you applying for? (Select one)

- Software Development Engineer (SDE)
- Machine Learning Engineer (MLE)
- Data Scientist (DS)

PART 2: DEMOGRAPHIC INFORMATION

4. Age:

5. Gender:

- Male
- Female
- Non-binary
- Prefer not to say

6. Race/Ethnicity (Please select all that apply):

- Arabic
- Black or African American
- East Asian
- Hispanic or Latino
- Native American
- Native Hawaiian or Other Pacific Islander
- South Asian
- White
- Other
- Prefer not to say

PART 3: EDUCATIONAL BACKGROUND

7. What is your current or highest level of education?

- Year 1 Undergraduate
- Year 2 Undergraduate
- Year 3 Undergraduate
- Year 4 Undergraduate
- Master's Student
- PhD Student
- Bachelor's Graduate (not current student)
- Master's Graduate (not current student)
- PhD Graduate

8. University/Institution Name:

9. Major(s):

10. Graduation Year / Expected Graduation Year:

11. How would you rate your overall academic performance in courses most relevant to your selected role?

- Excellent (Top 10%)
- Good (Top 10%-30%)
- Average (Top 30%-60%)
- Fair (Below Top 60%)

PART 4: ENGLISH PROFICIENCY

12. Is English your native language? (Yes / No)

13. (If No) Standardized English test scores (if applicable):
    - TOEFL
    - IELTS
    - Duolingo
    - CET-4
    - CET-6
    - Other
    - I have not taken any

PART 5: TECHNICAL & PROFESSIONAL EXPERIENCE

14. Number of relevant internships or full-time jobs:
    - 0
    - 1
    - 2
    - 3 or more

15. Brief description of most relevant work experience:

16. Have you published any peer-reviewed research papers? (Yes / No)

17. (If Yes) List of significant publications or link to academic profile:

18. Have you completed any significant personal/open-source projects? (Yes / No)

19. (If Yes) Link or description of the project you are most proud of:

20. Frequency of recent programming tasks WITHOUT AI assistance:
    - Daily
    - A few times a week
    - A few times a month
    - Rarely
    - Almost never

PART 6: FAMILIARITY WITH DEVELOPMENT ENVIRONMENTS & AI TOOLS

21. Primarily used IDEs or code editors (Select all that apply):
    - Visual Studio Code (VS Code)
    - JetBrains IDEs (e.g., PyCharm, IntelliJ)
    - Vim / Neovim
    - Jupyter Notebook / JupyterLab
    - Other

22. On a scale of 1 to 5 (1 = Novice, 5 = Expert), please rate your proficiency with Visual Studio Code (VS Code):

23. On a scale of 1 to 5 (1 = Not familiar at all, 5 = Very familiar), please rate your familiarity with container-based development or cloud-based IDEs:

24. On a scale of 1 to 5 (1 = Never, 5 = Almost always), please rate your frequency of relying on AI-powered coding assistants in your daily workflow:

25. Usage of GitHub Copilot specifically:
    - I use it daily as my primary AI assistant
    - I use it frequently (a few times a week)
    - I use it occasionally
    - I have tried it but do not use it regularly
    - I have never used it

26. Other AI coding tools used:

## J.3 Post-Test Questionnaire

*The following questionnaire was administered after participants completed all tasks to collect subjective feedback.*

### Part 1: Overall Experience & Usability

1. On a scale of 1 (Strongly Disagree) to 5 (Strongly Agree), please rate your agreement with the following statements:
   - The GitHub Codespaces environment was stable and easy to use.
   - The instructions in the README.md for each task were clear.
   - The submission process (./scripts/submit.sh) was straightforward.
2. Did you encounter any significant technical issues or confusion? (Open-ended response)

### Part 2: Comparison of Conditions (Human-Only vs. Human-AI)

3. Compared to tasks WITHOUT AI, how did tasks WITH AI affect your:
   - **Problem-Solving Speed:** (Much Slower / Slower / About the Same / Faster / Much Faster)
   - **Final Solution Quality/Correctness:** (Much Lower / Lower / About the Same / Higher / Much Higher)
4. On a scale of 1 to 5 (1 = Very Low, 5 = Very High), please rate the **Mental Effort (Cognitive Load)** for each condition:
   - Human-Only Condition:
   - Human-AI Collaboration Condition:
5. On a scale of 1 to 5 (1 = Not Confident at All, 5 = Very Confident), please rate your **Confidence** in your solution for each condition:
   - Human-Only Condition:
   - Human-AI Collaboration Condition:
6. In the Human-AI condition, which of the following roles did the AI play during your problem-solving process? (Select all that apply):
   - Brainstorming or exploring different solution strategies
   - Suggesting a fundamentally different approach or algorithm (including a change in the core algorithmic logic, different architectures, and the use of a completely different key library or tool)
   - Explaining high-level concepts or design trade-offs
   - Generating boilerplate, repetitive, or utility code
   - Implementing a specific, well-defined function or logic
   - Debugging errors in my code
   - Refactoring or optimizing existing code
   - Other
7. On a scale of 1 (Strongly Disagree) to 5 (Strongly Agree), please rate your agreement with the following statements about the AI assistant:
   - I trusted the code suggestions provided by the AI assistant.
   - I felt the explanations from the AI assistant were reliable.

### Part 3: Order Effects

8. On a scale of 1 (Strongly Disagree) to 5 (Strongly Agree), please rate your agreement with the following statements:
   - My performance in later tasks was influenced by the tasks I completed earlier.
   - My strategy for Human-Only tasks was affected by my experience in Human-AI tasks.
   - My strategy for Human-AI tasks was affected by my experience in Human-Only tasks.
9. If you felt there was an influence, please briefly describe it. (Open-ended response)

PART 4: TASK & EVALUATION FEEDBACK

10. On a scale of 1 to 5 (1 = Not at all realistic, 5 = Very realistic), please rate how well the tasks reflected real-world programming challenges:

11. On a scale of 1 (Strongly Disagree) to 5 (Strongly Agree), please rate your agreement with the report's accuracy:

    - **Regarding Human-Only tasks:**
        - The **Functional Correctness** score accurately reflected my performance.
        - The **Efficiency Metrics** accurately reflected my effort.
    - **Regarding Human-AI Collaboration tasks:**
        - The **Functional Correctness** score accurately reflected my performance.
        - The **Interaction Cost** metrics accurately reflected my collaboration with the AI.

12. Please explain your ratings on the evaluation report's accuracy. (Open-ended response)

PART 5: FINAL OPEN-ENDED FEEDBACK

13. What was the most positive or satisfying part of your experience? (Open-ended response)

14. What was the most negative or frustrating part of your experience? (Open-ended response)

15. Do you have any other suggestions for improving HAI-Eval? (Open-ended response)

### J.4 EXPERIMENTAL PROTOCOL

This section outlines the full procedural workflow experienced by participants during a single session. It illustrates the evaluation process used in `HAI-Eval` for human developers and highlights the framework's ecological validity. The entire protocol is designed as a self-contained experience, with all tasks performed within a pre-configured, cloud-based development environment.

**Step 1: Environment Initialization.** Participants begin by launching a dedicated Codespace instance via a provided link using their assigned GitHub account. The environment installs all necessary dependencies and extensions automatically. Once ready, a fully functional, browser-based VS Code workspace appears, as shown in Figure 6. The interface consists of three main components: a file explorer on the left, an integrated terminal at the bottom, and a central code editor. Environment configurations vary by condition. In the human-AI collaboration condition ($C_3$), GitHub Copilot is pre-installed and activated, with its icon visible in the Activity Bar. In the human-only condition ($H$), the extension is omitted entirely. This setup mirrors a typical cloud-based development workflow and requires no manual setup from the participant.

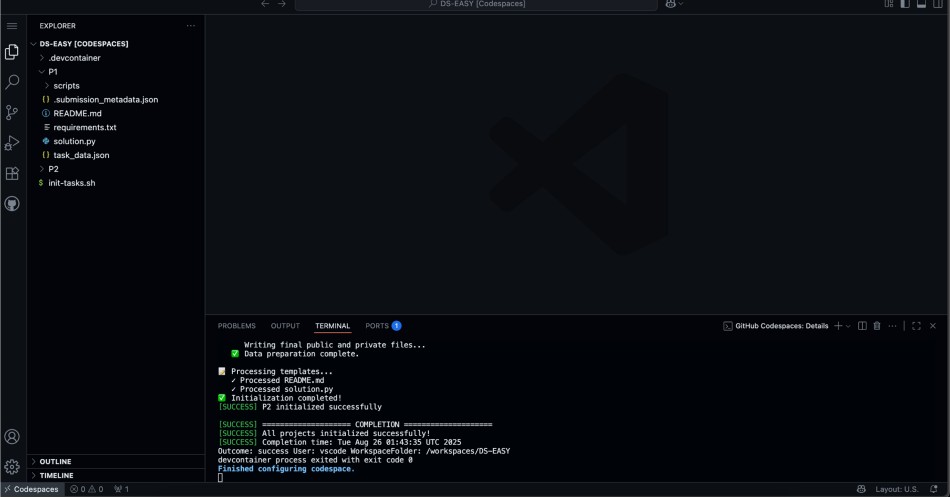

Figure 6: The standard workspace interface after initialization. The file explorer displays the core files for the task including `README.md` for the task description, `solution.py` for the participant's response, and scripts for submission. The terminal, which is used for executing code and submission scripts, indicates that all environments have been configured.

**Step 2: Task Comprehension.** Each task's requirements, scenario, dataset description, and objectives are documented in the corresponding `README.md` file. Participants are instructed to read this file carefully to understand the task context and expectations. Figure 7 illustrates the contents of a typical `README.md`.

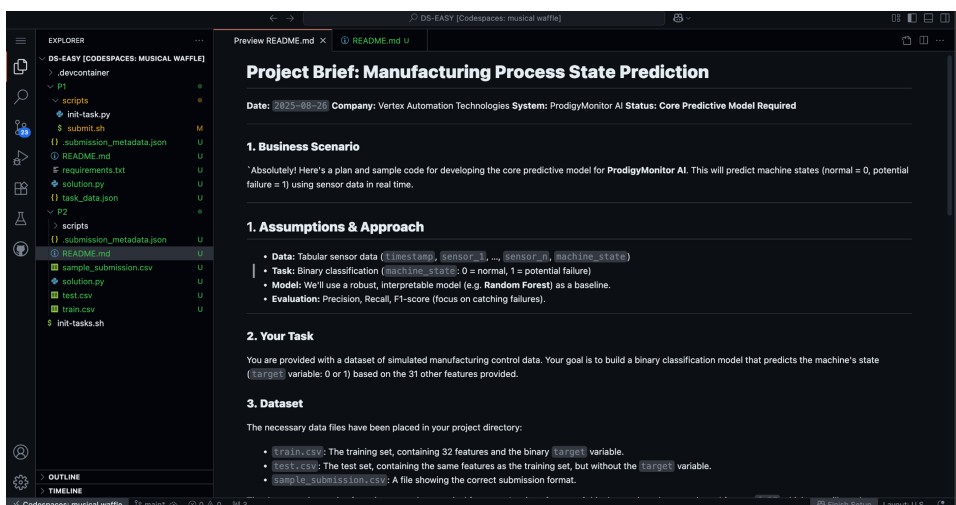

Figure 7: An example of a typical `README.md`.

**Step 3: Implementation.** Participants are required to write their code in the designated `solution.py` file to complete each task. For every task, we provide a starter code template that includes a basic framework and helper functions, allowing participants to focus on implementing the core logic. The total time limit for completing all tasks is two hours. Figure 8 shows an example of a starter code file.

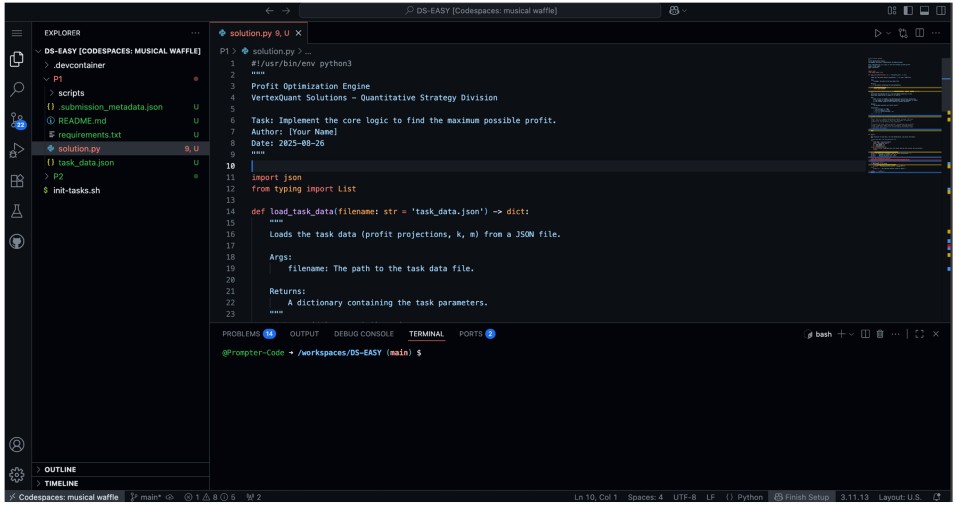

Figure 8: An example of a typical `solution.py`.

**Step 4: Solution Submission.** After completing their coding and local testing, participants are required to submit their solution by executing a shell script in the integrated terminal. They must first navigate to the corresponding problem directory and then run the submission command. This script automatically packages all necessary files and sends them to the backend evaluation server. The submission process is illustrated in Figure 9.

```
PROBLEMS 14    OUTPUT    DEBUG CONSOLE    TERMINAL    PORTS 2                                    bash - P1  + ∨  ⊓ 🗑 ⋯ | ⌞⌟ ×

● @Prompter-Code → /workspaces/DS-EASY (main) $ cd P1/
○ @Prompter-Code → /workspaces/DS-EASY/P1 (main) $ ./scripts/submit.sh

8 ⓘ 5  ⚠ 2                                    ◇ Prompter-Code (3 hours ago)  Ln 97, Col 5   Spaces: 4   UTF-8   LF   {} Shell Script   🖽 Finish Setup   Layout: U.S.   ⌸
```

Figure 9: An example of the submission process.

## K    DETAILED EXPERIMENTAL RESULTS

### K.1    DETAILED LLM BENCHMARKING RESULTS

Table 6 shows the performance comparison of SOTA LLMs across difficulty levels, and Table 7 shows the performance comparison across professional tracks. The results reveal a crucial insight consistent with our main findings. While there is a slight, expected degradation in performance as task difficulty increases, all models perform uniformly poorly across every difficulty level and every professional track. The pass rates for "Easy" tasks are nearly as low as those for "Hard" tasks, and performance shows no significant variation between different tracks.

This uniformity in failure strongly supports our finding that **the higher-order reasoning presents a fundamental wall**. It suggests that the primary bottleneck is neither the algorithmic complexity, which varies by difficulty, nor domain-specific knowledge, which varies by track, but rather the initial, higher-order challenge of requirement engineering and strategic planning inherent in HAI-Eval's "collaboration-necessary" design. Because the LLMs fundamentally struggle to interpret the context and formulate a valid plan for the problem, the subsequent difficulty or domain-specific knowledge of the implementation becomes largely irrelevant. This further validates that HAI-Eval effectively measures a fundamental capability gap that current LLMs cannot bridge, regardless of task difficulty or engineering domain.

Table 6: Detailed performance comparison of SOTA LLMs across difficulty levels.

| Metric | Claude-Sonnet-4 | | Claude-Sonnet-3.7 | | GPT-4.1 | | GPT-4o | | Gemini-2.5-Pro | |
|---|---|---|---|---|---|---|---|---|---|---|
| | $C_0$ | $C_{1\Delta(\%)}$ | $C_0$ | $C_{1\Delta(\%)}$ | $C_0$ | $C_{1\Delta(\%)}$ | $C_0$ | $C_{1\Delta(\%)}$ | $C_0$ | $C_{1\Delta(\%)}$ |
| **Easy** | | | | | | | | | | |
| Overall Pass@1 (%) | 1.33 | $4.00_{\uparrow 2.67}$ | 0.00 | $2.67_{\uparrow 2.67}$ | 0.00 | $2.67_{\uparrow 2.67}$ | 0.00 | $0.00_{-}$ | 0.67 | $4.00_{\uparrow 3.33}$ |
| Overall Pass@10 (%) | 6.67 | $7.33_{\uparrow 0.66}$ | 0.67 | $4.00_{\uparrow 3.33}$ | 2.67 | $5.33_{\uparrow 2.66}$ | 0.00 | $0.00_{-}$ | 1.33 | $4.00_{\uparrow 2.67}$ |
| Partial Pass@1 (%) | 27.63 | $43.29_{\uparrow 15.66}$ | 11.65 | $23.35_{\uparrow 11.70}$ | 13.39 | $40.59_{\uparrow 27.20}$ | 7.61 | $16.12_{\uparrow 8.51}$ | 14.71 | $34.73_{\uparrow 20.02}$ |
| Partial Pass@10 (%) | 33.89 | $45.82_{\uparrow 11.93}$ | 19.09 | $28.82_{\uparrow 9.73}$ | 16.78 | $40.92_{\uparrow 24.14}$ | 10.04 | $20.56_{\uparrow 10.52}$ | 17.24 | $38.33_{\uparrow 21.09}$ |
| **Medium** | | | | | | | | | | |
| Overall Pass@1 (%) | 0.67 | $2.67_{\uparrow 2.00}$ | 0.00 | $1.33_{\uparrow 1.33}$ | 0.00 | $2.00_{\uparrow 2.00}$ | 0.00 | $0.00_{-}$ | 0.00 | $1.33_{\uparrow 1.33}$ |
| Overall Pass@10 (%) | 2.67 | $3.33_{\uparrow 0.66}$ | 0.67 | $2.00_{\uparrow 1.33}$ | 1.33 | $3.33_{\uparrow 2.00}$ | 0.00 | $0.00_{-}$ | 0.67 | $1.33_{\uparrow 0.66}$ |
| Partial Pass@1 (%) | 18.50 | $29.29_{\uparrow 10.79}$ | 8.01 | $19.23_{\uparrow 11.22}$ | 11.88 | $20.27_{\uparrow 8.39}$ | 5.69 | $11.99_{\uparrow 6.30}$ | 6.17 | $14.96_{\uparrow 8.79}$ |
| Partial Pass@10 (%) | 20.66 | $32.11_{\uparrow 11.45}$ | 11.43 | $20.91_{\uparrow 9.48}$ | 14.36 | $21.98_{\uparrow 7.62}$ | 7.83 | $16.33_{\uparrow 8.50}$ | 8.09 | $15.90_{\uparrow 7.81}$ |
| **Hard** | | | | | | | | | | |
| Overall Pass@1 (%) | 0.00 | $2.00_{\uparrow 2.00}$ | 0.00 | $0.67_{\uparrow 0.67}$ | 0.00 | $0.67_{\uparrow 0.67}$ | 0.00 | $0.00_{-}$ | 0.00 | $1.33_{\uparrow 1.33}$ |
| Overall Pass@10 (%) | 1.33 | $2.00_{\uparrow 0.67}$ | 0.00 | $0.67_{\uparrow 0.67}$ | 0.00 | $2.00_{\uparrow 2.00}$ | 0.00 | $0.00_{-}$ | 0.00 | $1.33_{\uparrow 1.33}$ |
| Partial Pass@1 (%) | 11.63 | $18.07_{\uparrow 6.44}$ | 6.47 | $9.83_{\uparrow 3.36}$ | 8.20 | $10.08_{\uparrow 1.88}$ | 3.96 | $8.16_{\uparrow 4.20}$ | 3.92 | $14.32_{\uparrow 10.40}$ |
| Partial Pass@10 (%) | 9.74 | $24.37_{\uparrow 14.63}$ | 5.63 | $11.29_{\uparrow 5.66}$ | 10.77 | $11.56_{\uparrow 0.79}$ | 5.02 | $8.95_{\uparrow 3.93}$ | 5.90 | $15.22_{\uparrow 9.32}$ |

Table 7: Detailed performance comparison of SOTA LLMs across professional tracks.

| Metric | Claude-Sonnet-4 | | Claude-Sonnet-3.7 | | GPT-4.1 | | GPT-4o | | Gemini-2.5-Pro | |
|---|---|---|---|---|---|---|---|---|---|---|
| | $C_0$ | $C_{1\Delta(\%)}$ | $C_0$ | $C_{1\Delta(\%)}$ | $C_0$ | $C_{1\Delta(\%)}$ | $C_0$ | $C_{1\Delta(\%)}$ | $C_0$ | $C_{1\Delta(\%)}$ |
| **SDE** | | | | | | | | | | |
| Overall Pass@1 (%) | 0.00 | $3.33_{\uparrow 3.33}$ | 0.00 | $1.33_{\uparrow 1.33}$ | 0.00 | $1.33_{\uparrow 1.33}$ | 0.00 | $0.00_{-}$ | 0.00 | $2.67_{\uparrow 2.67}$ |
| Overall Pass@10 (%) | 4.00 | $4.00_{-}$ | 0.00 | $2.00_{\uparrow 2.00}$ | 1.33 | $1.33_{-}$ | 0.00 | $0.00_{-}$ | 0.67 | $2.67_{\uparrow 2.00}$ |
| Partial Pass@1 (%) | 18.81 | $27.00_{\uparrow 8.19}$ | 8.26 | $17.61_{\uparrow 9.35}$ | 13.48 | $25.32_{\uparrow 11.84}$ | 5.59 | $12.53_{\uparrow 6.94}$ | 7.48 | $21.45_{\uparrow 13.97}$ |
| Partial Pass@10 (%) | 22.83 | $34.12_{\uparrow 11.29}$ | 12.39 | $19.82_{\uparrow 7.43}$ | 14.14 | $26.26_{\uparrow 12.12}$ | 6.95 | $13.84_{\uparrow 6.89}$ | 11.23 | $24.25_{\uparrow 13.02}$ |
| **MLE** | | | | | | | | | | |
| Overall Pass@1 (%) | 1.33 | $2.67_{\uparrow 1.34}$ | 0.00 | $1.33_{\uparrow 1.33}$ | 0.00 | $1.33_{\uparrow 1.33}$ | 0.00 | $0.00_{-}$ | 0.00 | $1.33_{\uparrow 1.33}$ |
| Overall Pass@10 (%) | 3.33 | $4.67_{\uparrow 1.34}$ | 0.67 | $2.00_{\uparrow 1.33}$ | 0.67 | $1.33_{\uparrow 0.66}$ | 0.00 | $0.00_{-}$ | 0.67 | $2.00_{\uparrow 1.33}$ |
| Partial Pass@1 (%) | 22.19 | $31.97_{\uparrow 9.78}$ | 10.40 | $17.20_{\uparrow 6.80}$ | 9.60 | $22.80_{\uparrow 13.20}$ | 6.40 | $13.07_{\uparrow 6.67}$ | 8.40 | $22.57_{\uparrow 14.17}$ |
| Partial Pass@10 (%) | 23.01 | $34.72_{\uparrow 11.71}$ | 10.60 | $21.08_{\uparrow 10.48}$ | 13.93 | $24.41_{\uparrow 10.48}$ | 7.71 | $16.91_{\uparrow 9.20}$ | 10.51 | $22.61_{\uparrow 12.10}$ |
| **DS** | | | | | | | | | | |
| Overall Pass@1 (%) | 0.67 | $2.67_{\uparrow 2.00}$ | 0.00 | $2.00_{\uparrow 2.00}$ | 0.00 | $2.67_{\uparrow 2.67}$ | 0.00 | $0.00_{-}$ | 0.22 | $2.66_{\uparrow 2.44}$ |
| Overall Pass@10 (%) | 3.33 | $4.00_{\uparrow 0.67}$ | 0.67 | $2.67_{\uparrow 2.00}$ | 2.00 | $2.67_{\uparrow 0.67}$ | 0.00 | $0.00_{-}$ | 0.67 | $2.00_{\uparrow 1.33}$ |
| Partial Pass@1 (%) | 16.72 | $31.00_{\uparrow 14.28}$ | 7.47 | $17.60_{\uparrow 10.13}$ | 10.40 | $22.80_{\uparrow 12.40}$ | 5.47 | $10.67_{\uparrow 5.20}$ | 8.93 | $19.97_{\uparrow 11.04}$ |
| Partial Pass@10 (%) | 18.45 | $33.46_{\uparrow 15.01}$ | 13.16 | $20.12_{\uparrow 6.96}$ | 13.84 | $23.79_{\uparrow 9.95}$ | 8.23 | $15.09_{\uparrow 6.86}$ | 9.49 | $22.59_{\uparrow 13.10}$ |

## K.2 DETAILED HUMAN STUDY RESULTS

Table 8: Performance comparison of 4 conditions across professional tracks.

| SDE | $C_H$ | $C_0$ | $C_1$ | $C_2$ |
|---|---|---|---|---|
| **Overall Pass@1 (%)** | 20.00 | 0.00 | 3.33 | **30.00** |
| **Partial Pass@1 (%)** | 34.00 | 18.93 | 30.32 | **50.80** |
| **Completion Time (s)** | 2966 | 216 | 229 | **2567** |
| **Tokens (M)** | 0 | 0.47 | 0.51 | **2.18** |

| MLE | $C_H$ | $C_0$ | $C_1$ | $C_2$ |
|---|---|---|---|---|
| **Overall Pass@1 (%)** | 20.00 | 1.33 | 2.67 | **33.33** |
| **Partial Pass@1 (%)** | 33.20 | 22.19 | 31.97 | **50.33** |
| **Completion Time (s)** | 2791 | 203 | 214 | **2652** |
| **Tokens (M)** | 0 | 0.40 | 0.47 | **2.26** |

| DS | $C_H$ | $C_0$ | $C_1$ | $C_2$ |
|---|---|---|---|---|
| **Overall Pass@1 (%)** | 16.67 | 0.67 | 2.67 | **30.00** |
| **Partial Pass@1 (%)** | 34.40 | 16.72 | 31.28 | **49.73** |
| **Completion Time (s)** | 2603 | 233 | 262 | **2551** |
| **Tokens (M)** | 0 | 0.62 | 0.58 | **2.17** |

Table 8 breaks down performance by professional track, confirming our core findings are highly consistent across the three tracks. This stable performance across different expert user groups not only proves the generalizability of our conclusions but also validates the reasonable design and calibration of HAI-Eval's task templates. The tasks consistently pose appropriate challenges to experts from various fields while effectively limiting standalone AI performance at near-zero levels. This successfully creates a reliable problem space to precisely isolate and measure the core human-AI synergy within the emergent "co-reasoning partnership".

## K.3 DETAILED HUMAN FEEDBACK STATISTICS

This section provides selected statistics from the post-test questionnaire in Appendix J.3, organized according to its first three parts. The fourth part is provided in Section 5.2.

**Part 1: Ecological Validity of IDE.** To validate the **Ecological Validity** of our experimental setup, participants rated three core aspects of the user experience on a 5-point Likert scale (1 = Strongly Disagree, 5 = Strongly Agree). As shown in Figure 10, the feedback was consistently positive. The high mean scores for the usability of the IDE, the clarity of instructions, and the simplicity of the submission process confirm that our standardized environment provides a realistic development experience, ensuring that our experimental results reliably reflect participants' problem-solving capabilities.

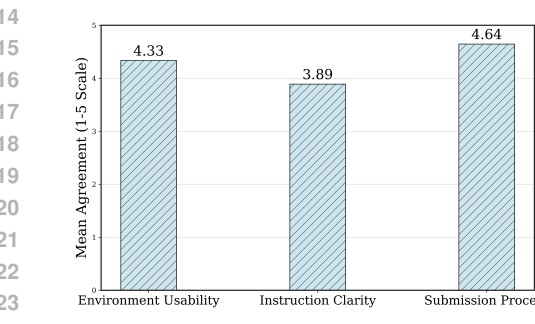
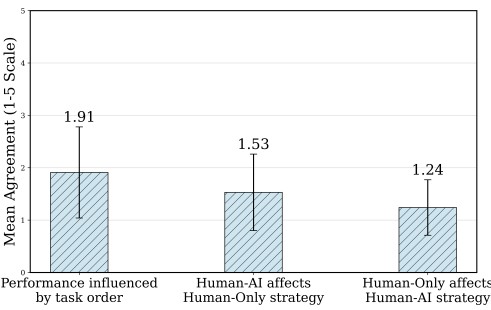

Figure 10: Averaged Ratings on overall user experience and usability.

Figure 11: Averaged ratings on order effects and standard deviation.

**Part 2.1: Confidence, Cognitive Load, & Trust.** Table 9 presents the descriptive statistics for four key subjective metrics presented in Section 5.2 and Figure 5. The results strongly support the core insights discussed in the main text. First, the data reveals a significant increase in participant confidence that is not met with a correspondingly large decrease in cognitive load, providing clear quantitative evidence for the "cognitive shift". Second, the high mean scores for both Trust in Code Suggestions and Trust in Explanations demonstrate a deep and holistic trust in the AI. The fact that trust in the AI's reasoning capabilities is rated as highly as its implementation output is powerful evidence for the "co-reasoning partnership".

Table 9: Descriptive statistics for confidence, cognitive Load, & trust).

| Metric | Mean | Standard Deviation |
|---|---|---|
| $\Delta$Confidence | 1.60 | 0.75 |
| $\Delta$Cognitive Ease | 0.51 | 1.66 |
| Trust in Code Suggestions | 4.09 | 0.90 |
| Trust in Explanations | 4.22 | 0.79 |

**Part 2.2: Coding Agent Usage.** Table 10 provides a detailed breakdown of the AI usage patterns reported by participants, comparing all 45 participants with the top-15 performers. The data shows that while implementation roles like "Generate boilerplate, repetitive, or utility code" and "Debug errors in my code" are nearly universally adopted, a significant majority also leveraged the AI for strategic tasks such as "Brainstorm or explore different solution strategies". The most notable distinction lies in high-level strategic usage: a remarkable 80% of top performers report using the AI to "Suggest a fundamentally different approach", compared to only 51% of the total participant group. These statistics provide the quantitative backing for the conclusion drawn in the main text: that high performance is strongly correlated with utilizing the AI for its advanced strategic capabilities.

Table 10: Statistics on AI usages reported by participants, comparing all 45 participants against the top 15 performers.

| AI Usage | #Users (total) | #Users (top-15) |
|---|---|---|
| Brainstorm or explore different solution strategies | 36 | 14 |
| Suggest a fundamentally different approach or algorithm | 23 | 12 |
| Explain high-level concepts or design trade-offs | 33 | 10 |
| Generate boilerplate, repetitive, or utility code | 45 | 15 |
| Implement a specific, well-defined function or logic | 39 | 13 |
| Debug errors in my code | 41 | 15 |
| Refactor or optimize existing code | 29 | 6 |

**Part 3: Order Effect.** To assess potential order effects within our fully counterbalanced, within-subject design, participants rated their agreement with three statements on the same Likert scale. The results are summarized in Figure 11. The low mean scores across all three statements indicate that participants did not perceive significant order effects, validating the robustness of our design.

## L  CASE STUDY

This case study, simplified from an existing trial in our user study, exemplifies our Observation 1-3 in Section 5.2. It demonstrates how human intervention resolves intentional "AI-Incomplete" barriers and how AI help human with both reasoning and productivity.

Notably, the selected participant in this case study is a male native Chinese speaker with high English proficiency (TOEFL Total 106/120; Reading 30/30; Writing 26/30). Despite his ability to comprehend the English task documentation successfully, the participant chose to interact with the agent primarily in Chinese to minimize cognitive load during the complex reasoning process (Van Rinsveld et al., 2015; Jouravlev et al., 2021). For clarity and readability, the multi-turn interaction logs below have been translated into English and summarized to highlight the core logical exchanges.

### L.1  TASK INTRODUCTION

**Task Description:** The problem is a variant of the classic algorithmic game Clickomania. The goal is to eliminate matching pairs from a grid, triggering gravity-like shifts that may create new matches, and to be efficient enough to pass all test cases within the time limit, not requiring a global optimal solution. We wrap this core algorithm with two real-world barriers:

➠ **Underspecified Requirements:** The rule for "column merging", what happens when a column becomes empty, is intentionally vague in the text. The documentation simply states: *Gravity behaves consistently with the provided legacy prototype.* This forces the solver to perform requirement engineering, realizing that an empty column must trigger a horizontal shift, a logic implicit only in the legacy code.

➠ **Legacy Code Distraction:** The provided "legacy prototype" is a slow, recursive Python script. It serves as a distractor, implementing a brute-force search that is functionally correct but computationally intractable for the target dataset.

### L.2  USER INTERACTION PROCESS

**AI-based Initial Effort.**    The process began with the participant attempting to generate the solution using the agent solely. Without human guidance, the agent (driven by Claude-4-Sonnet) failed on two fronts. First, distracted by the legacy code, it attempted to mimic the Brute-force Backtracking approach, resulting in a Time Limit Exceeded (TLE) error. Second, it struggled to extract the precise column-merging logic from the underspecified text, leading to failures in handling corner cases in the gravity logic.

**First-round Human Intervention.**    After the initial failure, the participant intervened the process. Firstly, after a multi-turn natural language dialogue analyzing the task constraints, he correctly cut through the legacy code distraction and identified that a global search was unnecessary, proposing a Greedy strategy. Due to the prompt is originally in Chinese and multi-turn, we provide a translated and summarized prompt below:

> **User:** *Ignore the recursive logic in the legacy script!!! You must use a greedy approach! Just loop through the whole board, find a pair that can be eliminated, remove it, and then repeat the loop until no more pairs exist.*

While the "Greedy" direction was theoretically correct, the implementation, Naive Iteration, implied a complexity of $O(N^3)$ or higher, which failed time constraints and related test cases.

**AI Co-Reasoning Correction.**    After attempting to follow the user's instruction to "loop through the board" and failing again, the agent proactively proposed a structural optimization. As the chain-of-thought of the agent was not completely presented, we polish and summarize the reply and action below:

> *While the greedy strategy is correct, a naive iteration (rescanning the whole board after every move) will be too slow. We should establish a **Candidate Queue**. When a pair is removed, we only add the specific rows and columns affected by the gravity shift back into the queue. This reduces the complexity.*

It represents the core observation defined in our study: LLMs did not merely code. They are also co-reasoning partners. The agent did not change the intent, but it provided a fundamentally different approach for execution, i.e., queue-based dynamic update.

**Final Human-AI Synergy.** Upon the participant's approval of the "Queue Strategy," the agent generated the implementation. This solution required approximately 200 lines of code, involving tedious logic for coordinate mapping, boundary checking, and gravity simulation. Notably, the AI-generated code handled the complex off-by-one errors and coordinate transformations effectively on the first try, a task that typically causes significant friction for human programmers.

The collaboration continued through the final stage. The participant reviewed the generated code, fixing minor domain-specific logic bugs and adjusting specific parameter configurations to ensure full alignment with the requirements. Most bugs were identified by the human participant, but corrected by the AI. This iterative process resulted in a final solution that passed all test cases, effectively demonstrating the complete synergy cycle.

## M  USE OF LLMS

We declare that LLMs are used as assistive tools during the preparation of this manuscript. Specifically, Gemini-2.5-Pro from Google and Claude-Sonnet-4 from Anthropic assist with polishing and translation. For the development of HAI-Eval, we use Claude-Sonnet-4 and Claude-Opus-4 as development tools. All final ideas, arguments, and implementations remain the responsibility of the human authors, who take full accountability for the entire content.

