# OpenReview forum: "HAI-Eval: Measuring Human-AI Synergy in Collaborative Coding"
_ICLR.cc/2026/Conference — ICLR 2026 Conference Desk Rejected Submission_

### Official Review · Reviewer_jwbk · 2025-10-30

**Soundness:** 3
**Presentation:** 3
**Contribution:** 4
**Rating:** 6
**Confidence:** 3

**Summary:**

The paper creates a method to create tasks for human-AI collaboration, an interface, and conducts a user study evaluating the performance of collaboration.

**Strengths:**

I find the user study and interface good contributions, but the most relevant contribution the approach the paper takes to the creation of tasks that are ecologically relevant but neither are solvable by humans nor LLMs alone. For me, this is the main contribution, and the rest is an evaluation of this method of creating tasks.

**Weaknesses:**

Some of the evaluations seem as if they are satisfied by construction. E.g., the fact that SOTA LLMs can't solve the tasks is in the specification of the task creation algorithm.

**Questions:**

- What part of the evaluations is "forced" by how you design the tasks?
- How similar to "real use" of LLMs are the tasks you are creating.
- [RealHumanEval](https://arxiv.org/abs/2404.02806) and the papers listed in [Centaur Evals](https://openreview.net/forum?id=LkdH35003E) could be additional helpful related literature.

---

> ### Author Response · Authors · 2025-11-20
> **Response to Reviewer jwbk’s Concerns (1/3)**
>
> We are sincerely grateful for your insightful review. We are encouraged by your positive assessment, especially in our contribution. Your constructive feedback provides us with a clear path to significantly strengthen our paper. We address each of your concerns below and have uploaded an updated manuscript that incorporates highlighted revisions based on your feedback.
>
> ### **W1/Q1: What part of the evaluations is "forced" by the design**
>
> Thank you for this insightful comment. You rightly point out that the low LLM performance is "in the specification" of our task creation. Our original use of the term "Finding 1" for this was misleading. We are grateful you helped us see this.
>
> To clarify this crucial point, we have updated our manuscript by replacing **“Finding”** with "**Observation**" (**Obs**). This systematic change is intended to precisely clarify the role of each point:
>
> * **Obs 1:** Serves as our methodological premise (“forced”).
> * **Obs 2 & Obs 3:** The core empirical findings (not 'forced'), quantifying the value and nature of human-AI collaboration after that premise is established.
>
> To answer Q1 directly, the "forced" part is **Obs 1**. We intentionally designed and validated our tasks to be intractable for standalone LLMs. This is a validation-driven observation and our methodological prerequisite.
>
> Our goal was not to simply prove that LLMs fail. It was to first establish this "**collaboration-necessary**" problem space, so that we could then empirically measure the true core findings of our evaluation:
>
> * **Obs 2 (Synergy Gain):** The performance lift for collaboration is a genuine measurement, not a constructed one.
> * **Obs 3 (Co-Reasoning Partnership):** The nature of this synergy, where humans shift from tool-users to "co-reasoning" partners, is an emergent behavior we discovered by analyzing participant feedback and logs.
>
> ### **Q2: How similar to "real use" of LLMs are the tasks**
>
> Continuing from W1/Q1, this question, how similar our "collaboration-necessary" tasks are to "real use", goes to one of our core design principles: **Ecological Validity**. It directly reflects its task requirement part: Tasks must simulate a real-world project context.
>
> We argue that our tasks are not only “real” but are specifically designed to be ecologically valid. We validate this from two perspectives:
>
> **1. Theoretical Grounding:**
> Our task design principles (**"AI-Incomplete" and "Human Reliance on AI"**) are rooted in established literature to ensure **Ecological Validity**. However, we admit that we did not make this theoretical grounding explicit enough in our submission. We are very grateful for your comment, which prompts us to strengthen the argumentation in our revised **Section 4.1**.
>
> We have added new discussions and citations in our revised manuscript. Specifically, for each of the individual strategies within both the "AI-Incomplete" and "Human Reliance on AI", we have now added corresponding citations. This shows that our task design is based on authentic challenges from real-world development, thus strengthening the theoretical foundation of our benchmark's Ecological Validity. Moreover, we provide a **reference list** for each cited paper, including their titles and usage. We hope this list demonstrates our theoretical grounding.
>
> Moreover, we are grateful for your excellent literature suggestions in Q3. The recommended "**RealHumanEval**" has been incorporated in the revised Section 4.1 as a powerful support to the strategy “underspecified requirements”.
>
> **2. Experimental Validation:**
> We also have direct experimental proof of Ecological Validity from our participants. We apologize that it was not prominent in our original submission due to page limit. The rebuttal rules now allow us to correct this.
>
> We have moved the participant feedback data on task realism from Appendix K.3 directly into the main text as **Section 5.3**. As this data shows, 45 participants strongly agreed that our tasks reflected "real-world challenges," giving it a high mean score of **4.07 out of 5.0**. This quantitative feedback is a powerful, direct validation.
>
> ### **Q3: Literature Recommendation**
>
> Thank you for these valuable references. We have added citations and discussions of both RealHumanEval, Centaur Evals, and papers listed in it to our revised manuscript.
>
> These newly introduced works are cited and discussed in the following sections: **Introduction**, **Related Works**, and **Section 4.1**. For your convenience, we provide a detailed list of added references and their usage in our next response. Please refer to it for details.
>
> **Once again, we truly appreciate your suggestions, which help us improve our paper significantly. We would be glad to discuss any remaining points or provide further clarification.**

---

> ### Author Response · Authors · 2025-11-20
> **Response to Reviewer jwbk’s Concerns: Added Reference List in Section 4.1 for Task Reality (2/3)**
>
> This response provides a list of the information of all new references cited in **Section 4.1** to address your concerns about task reality, including titles, the citation within the paper, the approximate locations, and the rationale for these citations. We hope this will help resolve your concerns. We also provide additional annotations for the references recommended by you.
>
> **Section 4.1-AI-Incomplete**
>
> * **"Detecting ambiguities in requirements documents using inspections"** (Kamsties et al., 2001): Establishes that requirement ambiguity is an inherent, fundamental challenge in software engineering requiring human clarification, confirming that our inclusion of "underspecified requirements" simulates real-world complexity rather than introducing artificial noise.
> * **(Recommended)** **"The RealHumanEval: Evaluating Large Language Models' Abilities to Support Programmers"** (Mozannar et al., 2024): Demonstrates the necessity of shifting evaluation focus from simple code generation to the ability to handle underspecified and realistic specifications, validating our core design philosophy for testing AI-incompleteness.
> * **"Why a diagram is (sometimes) worth ten thousand words"** (Larkin & Simon, 1987): Provides cognitive science evidence that parsing diagrams (e.g., UML) requires spatial indexing and logic reasoning, constituting a substantive multimodal and logic barrier for LLMs.
> * **"Unified modeling language: A complexity analysis"** (Siau & Cao, 2001): Quantifies the high cognitive load and information density of UML as a symbolic system, validating it as a rigorous test for an LLM's ability to extract logic from complex symbolic representations.
> * **"A survey on the practical use of UML for different software architecture viewpoints"** (Ozkaya & Erata, 2020): Confirms the persistent industrial relevance of formalized diagrammatic specifications, justifying the inclusion of UML/ER inputs as necessary representations of real-world "symbolic density" that impedes direct LLM comprehension.
> * **"SWE-bench: Can Language Models Resolve Real-World GitHub Issues?"** (Jimenez et al., 2024): Empirically demonstrates that LLMs struggle significantly with modifying large, complex legacy codebases, supporting our use of undocumented legacy code as a complexity factor that prevents autonomous LLM completion.
> * **"A survey on llm-based code generation for low-resource and domain-specific programming languages"** (Joel et al., 2024): Highlights the generalization deficits of general-purpose LLMs in domain-specific languages, providing a theoretical basis for embedding "domain-specific constraints" to challenge model adaptability.
> * **"On the effectiveness of large language models in domain-specific code generation"** (Gu et al., 2025): Provides recent experimental evidence that LLM performance degrades in domain-specific logic due to limited library proficiency, validating this dimension as an effective boundary between advanced reasoning and simple generation.
>
> **Section 4.1-Human Reliance on AI**
>
> * **"Human-AI collaboration: the effect of AI delegation on human task performance and task satisfaction"** (Hemmer et al., 2023): Establishes the theoretical basis for our "complementary strategy," showing that optimal task delegation naturally emerges when task design highlights AI capabilities relative to human limitations.
> * **"Transforming software development: Evaluating the efficiency and challenges of github copilot in real-world projects"** (Pandey et al., 2024): Empirically validates the efficiency dominance of AI in generating boilerplate code, confirming that our stacking of such tasks combined with time constraints effectively renders manual coding a non-viable, sub-optimal strategy.
> * **"Time pressure in software engineering: A systematic review"** (Kuutila et al., 2020): Identifies "time pressure" as a critical variable that alters developer behavior; we leverage this mechanism to force a strategic shift from inefficient manual methods to AI tool reliance.
> * **"Using an llm to help with code understanding"** (Nam et al., 2024): Indicates that developers primarily seek LLM assistance for unfamiliar concepts or APIs, confirming that our injection of "strategic knowledge gaps" (e.g., non-standard algorithms) effectively incentivizes tool usage as a specialized resource.

---

> ### Author Response · Authors · 2025-11-20
> **Response to Reviewer jwbk’s Concerns: Added Reference List in the Introduction and Related Works (3/3)**
>
> This response provides a list of the information of all new references cited in **Introduction** and **Related Works** to address your concerns, including titles, the citation within the paper, the approximate locations, and the rationale for these citations. We hope this will help resolve your concerns. We also provide additional annotations for the references recommended by you.
>
> **Section 1 (Introduction)**
>
> * **(Recommended)** **"Position: AI Should Not Be An Imitation Game: Centaur Evaluations"** (Haupt & Brynjolfsson, 2025): Critiques the field's focus on autonomous "imitation" benchmarks for only LLMs. We cite this to theoretically ground the critical need for evaluation frameworks that explicitly quantify human contribution and synergy, validating our shift away from purely autonomous metrics.
>
> **Section 2-User Studies in AI-Assisted Coding (Related Works)**
>
> * **(Recommended)** **"Evaluating human-language model interaction"** (Lee et al., 2022): Establishes a pioneering framework for assessing the interaction process across general tasks like creative writing. We use it to acknowledge the broader shift toward interactive evaluation while highlighting the specific lack of such standards for the rigorous logic and context requirements of coding.
> * **(Recommended)** **"Collaborative gym: A framework for enabling and evaluating human-agent collaboration"** (Shao et al., 2024): Proposes a general-purpose environment for measuring collaborative agents. We use it to acknowledge the broader shift toward collaborative evaluation while highlighting the specific lack of such standards for the rigorous logic and context requirements of coding.

---

### Official Review · Reviewer_NyJz · 2025-11-01

**Soundness:** 2
**Presentation:** 2
**Contribution:** 3
**Rating:** 2
**Confidence:** 4

**Summary:**

This paper introduces HAI-Eval, a benchmark for human-AI collaborative coding. They contribute a framework for generating items that are meant to be difficult for humans and AI in isolation, but that together, humans and AI are able to do better on the tasks. They contribute a standardized human eval protocol as well as an automated evaluation toolkit for LLMs, and perform experiments comparing LLM performance, human performance, and LLM + human performance.

**Strengths:**

This is a really important direction. Regarding originality, I am familiar with some work evaluating how AI might enhance human performance, but not a standardized benchmark for comparing human-AI teaming with just one of these factors. Trying to construct problems that expose the value of human-AI partnership is a unique approach that I think is conceptually very interesting. Regarding significance, the domain should broadly be of interest to many, as increasingly, human-AI teaming is the norm for software development. The paper is generally easy to understand, and considerable thought was put into evaluation construction as well as the human experiments.

**Weaknesses:**

While I think the approach of finding problems that are AI-incomplete but that are amenable to human reliance on AI for parts is a really interesting one, I’m left with a question: is the way that this is done meant to capture the real ways that humans and AI complement each other, and if so, does it accomplish this? The construction of this benchmark seems to rely on some intuitive building blocks for how humans and AI complement each other (e.g. humans providing clarification and decomposition, helping with diagrams), but also, could the way these items are constructed be a bit artificial? I have a hard time deciding, here, and would (ideally) want to understand what more in-the-wild data says about what’s effective in teaming.

Related, aspects of how the evaluation captures good partnership in a controlled way seems a bit opaque. Two related questions that seem quite light on important details in this paper. (1) How do you measure success when the task is ambiguous and requires human clarification? Might precisely how the human clarifies the task affect the difficulty of the task itself, if it’s left ambiguous ahead of time? And (2) how does the static LLM eval toolkit accomplish simulating interaction with a human user? I’m very confused by what that static evaluation is meant to measure, as 4.4 seems to say that it’s meant to simulate interaction with a human via HAI-EC (if so, how is HAI-EC engineered?), whereas in 5.1, HAI-EC is said to be used in C0. It would be really helpful to clarify these points.

The interpretations of Finding 1 in results seem unsupported. There is a claim that “(LLMs) are unable to perform higher-order reasoning tasks”, but the ways the problems have been constructed to be AI-incomplete (e.g. with incomplete instructions) don’t seem to support making such a claim — what’s higher-order reasoning, and what are the other ways (e.g. interpreting diagrams, clarifying) that lead to errors?

**Questions:**

Aside from the ones I couldn’t disentangle from the above weaknesses:

1. How are the C1 conditions chosen? How is this standardized? The presentation in the appendix makes it seem like there might be some variation in precisely how this is done.
2. How are the difficulty levels determined? How are they calibrated?
3. How do you establish when humans take a “fundamentally different approach”, as described in Finding 3?

---

> ### Author Response · Authors · 2025-11-20
> **Response to Reviewer NyJz’s Concerns (1/5)**
>
> We are grateful for your comprehensive and insightful review, which provides us with a clear path to significantly strengthen our paper. We address each of your concerns below and have uploaded an updated manuscript that incorporates highlighted revisions based on your feedback. For your reference, we also provide a **reference list** at the end of this rebuttal for the added citation in our paper, mainly focusing on addressing Weakness 1 & 3.
>
> ### **W1: Is the benchmark construction artificial?**
>
> We clarify that the constructed AI-Incompleteness of HAI-Eval, such as underspecified requirements and diagram interpretation, serve as ecologically valid operationalizations of established Software Engineering challenges. We have strengthened the manuscript by explicitly mapping these design choices to the empirical literature.
>
> We have revised **Section 4.1** to ground each AI-Incomplete and Human Reliance on AI strategy in specific literature. For your convenience, we provide titles and reasons of use for these papers in the **reference list**. Specifically, we detail the rationale for the two examples mentioned by you below:
>
> 1. **"Humans provide clarification and decomposition" (Addressing Ambiguity).** Our **Underspecified Requirements** strategy refers to the findings of Mozannar et al. **[1]** and Hemmat et al. **[2]**. Their empirical studies show that bridging the gap between vague intent and formal specification constitutes a primary cost in real-world AI-assisted coding. Barke et al. **[3]** further categorize this as the "Grounded Copilot" interaction mode where human decomposition acts as the essential mechanism for resolving task ambiguity. This confirms that our design models an authentic interaction pattern.
> 2. **"Helping with diagrams" (Multimodal Context).** The inclusion of UML/ER diagrams represents standard industrial practice rather than an artificial puzzle. Petre **[4]** demonstrates that professional developers rely heavily on diagrams to convey constraints often omitted in text. Furthermore, Larkin & Simon **[5]** establish that diagrammatic reasoning requires distinct cognitive inference processes, such as locational indexing, compared to text processing. This justifies its inclusion as a specific test of Human-AI Synergy.
>
> We also offer direct quantitative evidence from our user study. As detailed in our original **Appendix K.3 (Part 4)** , 45 expert participants rated the statement "Tasks reflected real-world challenges" with a mean score of **4.07/5.0**. This high agreement confirms that expert users perceive the task design as a simulation of the messy reality of professional engineering. We apologize that it was not prominent in our original submission due to space constraints. We have moved this analysis to the main text as **Section 5.3**.
> Regarding the in-the-wild data, while we agree on its value, we believe incorporating it here would divert from our primary contribution.
> The dynamics of in-the-wild teaming have been extensively explored in prior qualitative studies (discussed in Related Works). Thus, our specific goal is to address the lack of standardization in those studies by providing a controlled, reproducible benchmark. Relying on uncontrolled in-the-wild data would reintroduce the noise and variability HAI-Eval aims to eliminate.
> Instead, to address your concern regarding the realism of our collaboration patterns without sacrificing control, we have added a detailed **Case Study in Appendix L**. It concretely demonstrates the authentic, 'in-the-wild' nature of the challenges and synergy patterns captured within our standardized framework. If it still fails to address your concerns, we are glad to provide more information.
>
> **References:**
>
> [1] Mozannar, H., Bansal, G., Fourney, A., & Horvitz, E. (2024). Reading Between the Lines: Modeling User Behavior and Costs in AI-Assisted Programming. CHI '24.
>
> [2] Hemmat, A., et al. (2025). Research Directions for Using LLM in Software Requirement Engineering: A Systematic Review. Frontiers in Computer Science.
>
> [3] Barke, S., James, M. B., & Polikarpova, N. (2023). Grounded Copilot: How Programmers Interact with Code-Generating Models. OOPSLA.
>
> [4] Petre, M. (2013). UML in practice. ICSE '13.
>
> [5] Larkin, J. H., & Simon, H. A. (1987). Why a diagram is (sometimes) worth ten thousand words. Cognitive Science.

---

> ### Author Response · Authors · 2025-11-20
> **Response to Reviewer NyJz’s Concerns (2/5)**
>
> ### **W2.1: Clarification of Task Ambiguity**
>
> Thank you for this question. To directly answer it: in HAI-Eval, "ambiguity" does not mean the problem is open-ended. Drawing on established work in software engineering **[1]** (which is also cited in **Section 4.1-AI-Incomplete**), **it refers to underspecified requirements that serve as a cognitive barrier**—a fixed hurdle that humans must bridge through interpretation.
>
> * **"Clarification" is a Process of Convergence, Not Redefinition:** The "human clarification" we describe is a cognitive process of **converging** on the single intended logic hidden within the context. The participant must use higher-order reasoning skills (e.g., requirement engineering) to deduce specific constraints rather than inventing them. This simulates the professional reality where requirements are rarely perfect, placing the difficulty in the process of clarification itself.
> * **The Ground Truth is Deterministic:** Critically, while the problem formulation requires interpretation, the required solution metrics for any given task instance are deterministic and non-negotiable. Success is measured objectively by our evaluation system. Our Agentic Task System generates a fixed suite of hidden test cases alongside each task instance. **All submissions—whether from an AI or a human—are judged against the same suite of test cases.** Therefore, task difficulty remains constant. While the cognitive path to the solution may vary, the standard for correctness does not. This ensures that our evaluation is fair, standardized, and reproducible.
>
> ### **W2.2: Clarification of HAI-EC**
>
> We are grateful for the opportunity to clarify that HAI-EC is designed as an **automated evaluation controller**, not a simulator of human cognition or collaboration. We have revised **Section 4.4** to make this distinction clearer.
>
> When we state that HAI-EC "replicates the interaction pattern of a human developer," we refer strictly to the **mechanical workflow** within the IDE. We respectfully point out that our original manuscript explicitly defines this 'interaction pattern' in the sentences immediately following the one cited by the reviewer. This definition confirms it refers only to a mechanical workflow, not human simulation. HAI-EC automates the procedural steps a developer would take:
>
> * Opening the VS Code environment.
> * Invoking Copilot via the API.
> * Reading the README.
> * Running tests and iterating based on failure feedback.
>
> The purpose of HAI-EC is to provide a controlled, ecologically valid test harness for the C0 and C1 conditions. By automating the standard developer workflow, HAI-EC ensures that the AI agents operate within the **exact same environment and toolchain** as the human participants in the C2 condition. This eliminates confounding variables related to the environment and allows for a direct, fair comparison between autonomous AI performance and human-AI collaborative performance.
>
> **References:**
>
> [1] Kamsties, E., et al. (2001). Detecting ambiguities in requirements documents using inspections. WISE’01.

---

> ### Author Response · Authors · 2025-11-20
> **Response to Reviewer NyJz’s Concerns (3/5)**
>
> ### **W3: "Finding 1" Logic and Definition of Higher-Order Reasoning**
>
> We agree that framing the low LLM performance as "Finding 1" might appear circular given the AI-Incomplete design goal. We rename "**Finding 1**" to "**Observation 1**" (**Obs 1**) in **Section 5.2** to clarify that this result serves as a **methodological validation**. It first establishes our "collaboration-necessary" problem space, so that we could then empirically measure the true core findings of our evaluation, i.e., Obs 2 & 3 about the synergy.
>
> Additionally, we formalize the definition of "higher-order reasoning" using the renowned **Relational Complexity Theory [1]** in **Section 3 (Necessary Collaboration)** and **Section 4.1 (AI-Incomplete)**.
>
> To fully address your concern, we formalize higher-order reasoning in HAI-Eval as tasks requiring:
>
> 1. **Intent inference under ambiguity:** This requires recovering latent constraints not stated verbatim, corresponding to our Underspecified Requirements component. Errors occur because LLMs often default to high-probability patterns rather than verifying specific, unstated intent **[2]**.
> 2. **Symbolic logic extraction:** This involves binding variables across different modalities, which increases processing capacity load according to Relational Complexity Theory. This corresponds to our Multimodal Specifications. Crucially, this is not merely visual perception, but the processing of information-dense symbolic systems with high cognitive load **[3]**, constituting a substantive logic barrier rather than a simple perception task **[4]**.
> 3. **Global constraint verification:** This corresponds to undocumented legacy logic in an existing system or system-wide business logic in our design. This requires **delocalized planning [5]**, a recognized higher-order reasoning skill.
>
> **References:**
>
> [1] Halford, G. S., Wilson, W. H., & Phillips, S. (1998). Processing capacity defined by relational complexity: Implications for comparative, developmental, and cognitive psychology. Behavioral and Brain Sciences.
>
> [2] Mozannar, H., Bansal, G., Fourney, A., & Horvitz, E. (2024). Reading Between the Lines: Modeling User Behavior and Costs in AI-Assisted Programming. CHI '24.
>
> [3] Siau, K., & Cao, Q. (2001). Unified modeling language: A complexity analysis. Journal of Database Management (JDM), 12(1), 26-34.
>
> [4] Larkin, J. H., & Simon, H. A. (1987). Why a diagram is (sometimes) worth ten thousand words. Cognitive Science.
>
> [5] Letovsky, S., & Soloway, E. (1986). Delocalized plans and program comprehension. IEEE software, 3(3), 41.

---

> ### Author Response · Authors · 2025-11-20
> **Response to Reviewer NyJz’s Concerns (4/5)**
>
> ### **Q1: How are the C1 conditions chosen and how is this standardized?**
>
> We apologize if Appendix G gave an impression of variability. In fact, it serves as a **strict exclusion protocol** designed to eliminate variability.
>
> * **Permitted:** Only interventions for strictly defined "procedural failures" (e.g., environment setup, dependency issues) are allowed.
> * **Prohibited:** Absolutely no "logical or semantic assistance" is permitted.
>
> This standardization is crucial, as it removes confounding variables (environmental "noise") to ensure we are fairly measuring the upper bound of the LLM's "core logical reasoning capabilities“ without artificially boosting its performance.
>
> To further clarify this point, we have modified the title of **Appendix G** to **Standardized Intervention Protocol for Condition C1**.
>
> ### **Q2: How do you determine and calibrate the difficulty levels?**
>
> We apologize that the details regarding difficulty calibration were previously hard to locate within Appendix D.2. To rectify this, we have added a direct reference in **Section 4.1** of the main text to guide readers to the full methodology, and modified **Appendix D** to make it clear.
>
> As detailed in Appendix D, our difficulty levels are determined by a two-stage process:
>
> * **Definition:** We first defined difficulty based on a set of objective metrics, including "**Algorithmic Complexity**," "**Implementation Scope**," and "**Time Requirements**". They are grounded in existing software engineering education literature like **[1]**.
> * **Calibration:** These indicators are calibrated through a dual independent expert validation process. As reported in Appendix D.3, this process achieved an **Inter-Rater Reliability (IRR) score of 0.97**, demonstrating that our difficulty classification is highly consistent.
>
> ### **Q3: How do you establish when humans take a “fundamentally different approach”**
>
> Thank you for pointing out this methodological detail. You are correct that "fundamentally different approach" was not formally defined in the original text. To further clarify this, we have added illustrations in **Obs 3 in Section 5.2** and the **post-test questionnaire in Appendix J.3** of our revised manuscript. You may also refer to the added case study in **Appendix L** for an intuitive example.
>
> To directly answer your first point, the data is derived from participant feedback from question 6 of our post-test questionnaire in Appendix J.3. This self-reported data is not analyzed in isolation. It is **cross-validated** by an expert analysis of user logs.
>
> To answer your second point, we define this as a fundamental shift in the **core solution strategy**, which can manifest in several ways:
>
> * A change in the core algorithmic logic (e.g., from a brute-force approach to an optimized dynamic programming).
> * A different architectural approach (e.g., the AI suggesting a generator-based pipeline to process a large file lazily, rather than loading it all into memory at once).
> * The use of a completely different key library or tool that reshapes the entire implementation path (e.g., leveraging pandas for a task the user was attempting with manual list manipulation).
>
> **Reference:**
>
> [1] Pelánek, R., Effenberger, T., & Čechák, J. (2022). Complexity and difficulty of items in learning systems. International Journal of Artificial Intelligence in Education, 32(1), 196-232.
>
>
> **Once again, we truly appreciate your comments. We would be glad to discuss any remaining points or provide further clarification.**

---

> ### Author Response · Authors · 2025-11-20
> **Response to Reviewer NyJz’s Concerns (5/5) - Reference List**
>
> Below we list all new references with locations and rationales. Hope this will help resolve your concerns.
>
> **Section 3-Necessary Collaboration**
>
> * **"Processing capacity defined by relational complexity: Implications for comparative, developmental, and cognitive psychology"** (Halford et al., 1998): Establish a foundational framework where reasoning difficulty is defined by the complexity of relations between interacting variables; we cite this to substantiate that our implementation of "higher-order reasoning" (by interacting constraints like legacy code and domain logic) is scientifically grounded on crucial research on cognitive complexity.
>
> **Section 4.1-AI-Incomplete**
>
> * **"Detecting ambiguities in requirements documents using inspections"** (Kamsties et al., 2001): Define requirement ambiguity is an inherent challenge in software engineering requiring human clarification, confirming that our inclusion of "underspecified requirements" simulates real-world complexity.
> * **"The RealHumanEval: Evaluating Large Language Models' Abilities to Support Programmers"** (Mozannar et al., 2024): Validate the need to shift evaluation focus from simple code generation to the ability to handle underspecified and realistic specifications, validating our design for "underspecified requirements".
> * **"Why a diagram is (sometimes) worth ten thousand words"** (Larkin & Simon, 1987): Provide cognitive science evidence that parsing diagrams (e.g., UML) requires spatial indexing and logic reasoning, constituting a substantive multimodal and logic barrier for LLMs.
> * **"Unified modeling language: A complexity analysis"** (Siau & Cao, 2001): Quantify UML's high cognitive load/information density as a symbolic system, validating it as a rigorous test for LLMs’ ability to extract logic from complex symbolic representations.
> * **"A survey on the practical use of UML for different software architecture viewpoints"** (Ozkaya & Erata, 2020): Confirm the industrial relevance of formalized diagrams, justifying the inclusion of diagram inputs as necessary representations of real-world "symbolic density" that impedes direct LLM comprehension.
> * **"SWE-bench: Can Language Models Resolve Real-World GitHub Issues?"** (Jimenez et al., 2024): Demonstrate that LLMs struggle significantly with modifying large, complex legacy codebases, supporting our use of undocumented legacy code as a complexity factor that prevents autonomous LLM completion.
> * **"A survey on llm-based code generation for low-resource and domain-specific programming languages"** (Joel et al., 2024): Highlight the generalization deficits of general-purpose LLMs in domain-specific languages, providing a theoretical basis for embedding "domain-specific constraints" to challenge model adaptability.
> * **"On the effectiveness of large language models in domain-specific code generation"** (Gu et al., 2025): Provide recent experimental evidence that LLM performance degrades in domain-specific logic due to limited library proficiency, validating this dimension as an effective boundary between advanced reasoning and simple generation.
>
> **Section 4.1-Human Reliance on AI**
>
> * **"Human-AI collaboration: the effect of AI delegation on human task performance and task satisfaction"** (Hemmer et al., 2023): Establish the theoretical basis for our "complementary strategy," showing that optimal task delegation naturally emerges when task design highlights AI capabilities relative to human limitations.
> * **"Transforming software development: Evaluating the efficiency and challenges of github copilot in real-world projects"** (Pandey et al., 2024): Empirically validate the efficiency dominance of AI in generating boilerplate code, confirming that our stacking of such tasks combined with time constraints effectively renders manual coding a non-viable, sub-optimal strategy.
> * **"Time pressure in software engineering: A systematic review"** (Kuutila et al., 2020): Identify "time pressure" as a critical variable that alters developer behavior; we leverage this mechanism to force a strategic shift from inefficient manual methods to AI tool reliance.
> * **"Using an llm to help with code understanding"** (Nam et al., 2024): Indicate that developers primarily seek LLM assistance for unfamiliar concepts or APIs, confirming that our injection of "strategic knowledge gaps" (e.g., non-standard algorithms) effectively incentivizes tool usage as a specialized resource.
>
> **Appendix D.2-Difficulty Calibration and Quality Criteria**
>
> * **"Complexity and difficulty of items in learning systems"** (Pelánek et al., 2022): Provide a systematic framework distinguishing between intrinsic task complexity and empirical difficulty; we derive our "Difficulty Consistency" metric definitions—specifically the mapping of algorithmic depth, scope, and time to difficulty levels—directly from their analysis of structural complexity indicators in programming tasks.

---

### Official Review · Reviewer_eTTX · 2025-11-01

**Soundness:** 3
**Presentation:** 3
**Contribution:** 3
**Rating:** 6
**Confidence:** 5

**Summary:**

The authors propose a new benchmark system that evaluates AI coding assistants in vivo. They validate this benchmark by showing that the results on this benchmark differ dramatically from those of a human coder alone or an AI coder alone.

**Strengths:**

- I like the idea. It does indeed identify a key gap in existing benchmarks. It also constructs the benchmark in a way that fits a lot with my own experiences on where Agentic AI coding systems are most useful.

**Weaknesses:**

- I think my biggest criticism (which is a relatively small one) is that the sample of participants is very biased. This should be mentioned in the main text. I think it would be sufficient to note the biggest biases in the sample in the main text, namely that all the participants identified as East Asian and that all of the participants regularly use AI coding assistants (this would just imply to the reader that some care needs to be taken when drawing conclusions). It should also be noted (though in the appendix is fine) whether participants were recruited among personal acquaintances of the authors. This isn't a dealbreaker, but should be noted.
- It's not completely true that other assessments assume a perfectly defined problem. I would add the clause "most" into that. This is false in two ways: (1) LLM-based evaluations (e.g., LLM-as-a-Judge) allow more soft definitions of a task, and (2) advanced versions of that (e.g., Agent-as-a-Judge) allow one to evaluate on much higher-level definitions of the problem, rendering the "perfectly defined problem" not a limitation. It would be worth mentioning these kinds of evaluation strategies here.
- Figure 2 could be cleaned up a bit. I think you could simplify this quite a bit. I find in-figure sentences are useful in annotating what's going on there.
- What is this Anti-LLM? It's only mentioned in the conclusion and a figure. Maybe avoid the new notation if it's not really used?
- I'm not a fan of this "commitment to maintain" that's being done a lot nowadays. This is near impossible to guarantee in academia (and will be embarrassing for the authors when someone reads the paper 10 years from now and this benchmark has reached its end-of-life). A more useful commitment would be to ensure that others can easily compute the metrics themselves. Maybe just mention that you are releasing the source code instead?

**Questions:**

See Weaknesses.

---

> ### Author Response · Authors · 2025-11-20
> **Response to Reviewer eTTX’s Concerns**
>
> We are incredibly grateful for your thorough and insightful review. Your positive assessment is a great encouragement to us. We have carefully considered all your constructive suggestions and believe your feedback has significantly helped us improve our paper. We address each of your concerns below and have uploaded an updated manuscript that incorporates highlighted revisions based on your feedback.
>
> ### **W1: Sample Bias Notification**
>
> Thank you for the constructive advice. We completely agree that these details are important to state clearly in the main text.
>
> As you noted, the details were previously presented in Appendix A and H. To address your concerns, in our revised paper, we move all critical details directly into the **‘Participants’ of Section 5.1** in the main text. This paragraph now explicitly states:
>
> * The recruitment method (including "personal contacts"), moved from Appendix H.1.
> * All 45 participants were identified as East Asian and required to be regular users of coding agents.
> * A concluding sentence, as you suggested, cautioning that "some care needs to be taken when drawing conclusions" from this specific demographic.
>
> ### **W2: Perfectly Defined Problem Assumption**
>
> We agree that advanced assessment methods (LLM/Agent-as-a-Judge) are powerful tools for evaluating tasks that are not perfectly defined. Based on your advice, we revise the Introduction, adding "most" to our claim and a discussion (with citations) to the **2nd paragraph of the Introduction**. This discussion acknowledges that while these advanced assessment methods exist, the datasets with “imperfectly defined problems” to properly support them (like HAI-Eval) have been lacking.
>
> We also believe these methods are a powerful complement to our work. As already noted in our Future Work (Appendix A), we plan to integrate these advanced LLM-based assessments to evaluate qualitative code metrics, perfectly aligning with your suggestion.
>
> ### **W3: Revise Figure 2**
>
> We update Figure 2 in the revised paper. Following your suggestion, we simplify the figure and add in-figure sentences to make the architecture more intuitive. We hope this version addresses your concern.
>
> ### **W4: Revise “Anti-LLM”**
>
> Thank you for the professional advice. For the two instances of "Anti-LLM", we revise them individually based on the context.
> * In **Figure 1**, we replace it with our established terminology, "**collaboration-necessary**," to maintain consistency.
> * In the **Conclusion**, we remove the term entirely, as the subsequent phrase already specifies that the tasks "necessitate human-AI collaboration," thus avoiding redundancy.
>
> ### **W5: Commitment to Maintain**
>
> Thank you for this kind and pragmatic suggestion. We agree our original phrasing was overly idealistic.
>
> Following your advice, we revise the paper to emphasize reproducibility and code release. Specifically, we update **Sections 4.1 and 4.4** to highlight open-source availability and independent reproducibility, while also replacing the vague 'long-term maintenance' commitment.
>
> Moreover, we would like to clarify that while we remove the 'long-term' commitment, we do have a concrete maintenance plan for the near term. This plan is highly motivated by HAI-Eval's central role in our ongoing project, which deploys adapted versions in application scenarios. We adopt a balanced approach that we believe addresses your concerns about over-promising while reflecting our active work. This approach includes the following steps:
>
> * **Empower Community Contribution:** We revise and reposition our Repository Governance (**Appendix E.1**, original Appendix E.3) to underscore the significant value of the open-source community and encourage greater participation in the project's maintenance and development. To ensure the traceability of contributions and the fairness of evaluations, we will manage dataset changes through strict versioning and clearly annotate results on our leaderboard with corresponding benchmark versions.
> * **Refine Maintenance Protocol:** Following your suggestion, we adopt a more actionable maintenance approach. First, we remove the commitment to periodic, holistic maintenance from the Maintenance Protocol (original Appendix E.1). We will conduct quarterly evaluations of new SOTA models in Copilot agent mode and provide a continuously updating leaderboard on GitHub for one year. We retain the part related to template validation and static dataset synchronization, redefining them as "Maintenance Process” for contributions to shift the focus towards community-driven maintenance.
> * **Remove Expansion Protocol:** We remove the Expansion Protocol (original Appendix E.2). Ambitious ideas, including new programming languages and professional tracks, are moved to **Future Work (Appendix A)**.
>
> **Once again, we truly appreciate your suggestions, which help us improve our paper significantly. We would be glad to discuss any remaining points or provide further clarification.**

---

### Author Response · Authors · 2025-11-28
**Looking Forward to Further Discussion**

Dear Reviewers,

Wishing you having a happy and blessed Thanksgiving!

We sincerely appreciate the time and effort you have invested in reviewing our manuscript. We have carefully addressed all the concerns raised in your initial reviews and provided detailed responses along with a revised version of our paper.

As the discussion period is underway, we would be grateful if you could take a moment to review our response and share any further thoughts or questions. We remain committed to engaging constructively in this dialogue to enhance the quality of our work.

Thank you for your continued attention.

Best regards, Authors

---

### Note · Program_Chairs · 2026-01-17
**Submission Desk Rejected by Program Chairs**

The following references in this submission do not refer to real documents and/or have major errors in bibliographic information:

 LiveCodeBench: A comprehensive benchmark for general-purpose language agents